# TTS-VAR: A Test-Time Scaling Framework for Visual Auto-Regressive Generation

Zhekai Chen[1]   Ruihang Chu[2*]   Yukang Chen[3]   Shiwei Zhang[2]   Yujie Wei[2]
Yingya Zhang[2]   Xihui Liu[1*]

[1] HKU MMLab      [2] Tongyi Lab, Alibaba Group      [3] CUHK
zkchen66@outlook.com

## Abstract

Scaling visual generation models is essential for real-world content creation, yet requires substantial training and computational expenses. Alternatively, test-time scaling has garnered growing attention due to resource efficiency and promising performance. In this work, we present *TTS-VAR*, the first general test-time scaling framework for visual auto-regressive (VAR) models, modeling the generation process as a path searching problem. To dynamically balance computational efficiency with exploration capacity, we first introduce an adaptive descending batch size schedule throughout the causal generation process. Besides, inspired by VAR's hierarchical coarse-to-fine multi-scale generation, our framework integrates two key components: (i) At coarse scales, we observe that generated tokens are hard for evaluation, possibly leading to erroneous acceptance of inferior samples or rejection of superior samples. Noticing that the coarse scales contain sufficient structural information, we propose clustering-based diversity search. It preserves structural variety through semantic feature clustering, enabling later selection on samples with higher potential. (ii) In fine scales, resampling-based potential selection prioritizes promising candidates using potential scores, which are defined as reward functions incorporating multi-scale generation history. Experiments on the powerful VAR model Infinity2B show a notable 8.7% GenEval score improvement (0.69→0.75). Key insights reveal that early-stage structural features effectively influence final quality, and resampling efficacy varies across generation scales. Code is available at https://github.com/ali-vilab/TTS-VAR.

## 1 Introduction

Recent years have witnessed significant progress in image generative models [1–4]. Previous text-to-image generative models primarily rely on diffusion models [5, 6], which iteratively denoise the latent to generate high-quality images from random noise. Yet, advancements in large language models (LLMs) [7–9] have spurred interest in Auto-Regressive (AR) architectures for image generation [10–12], leveraging sequential modeling to capture visual patterns. Among these, Visual Auto-Regressive Modeling (VAR) [13, 14] has emerged as a groundbreaking paradigm. It encodes images into multi-scale coarse-to-fine representations and progressively predicts the "next scale" to synthesize images through hierarchical aggregation. Due to its superior efficiency and the potential for unified integration with LLMs, VAR is fast emerging as a key research frontier.

---

* Corresponding Authors

39th Conference on Neural Information Processing Systems (NeurIPS 2025).

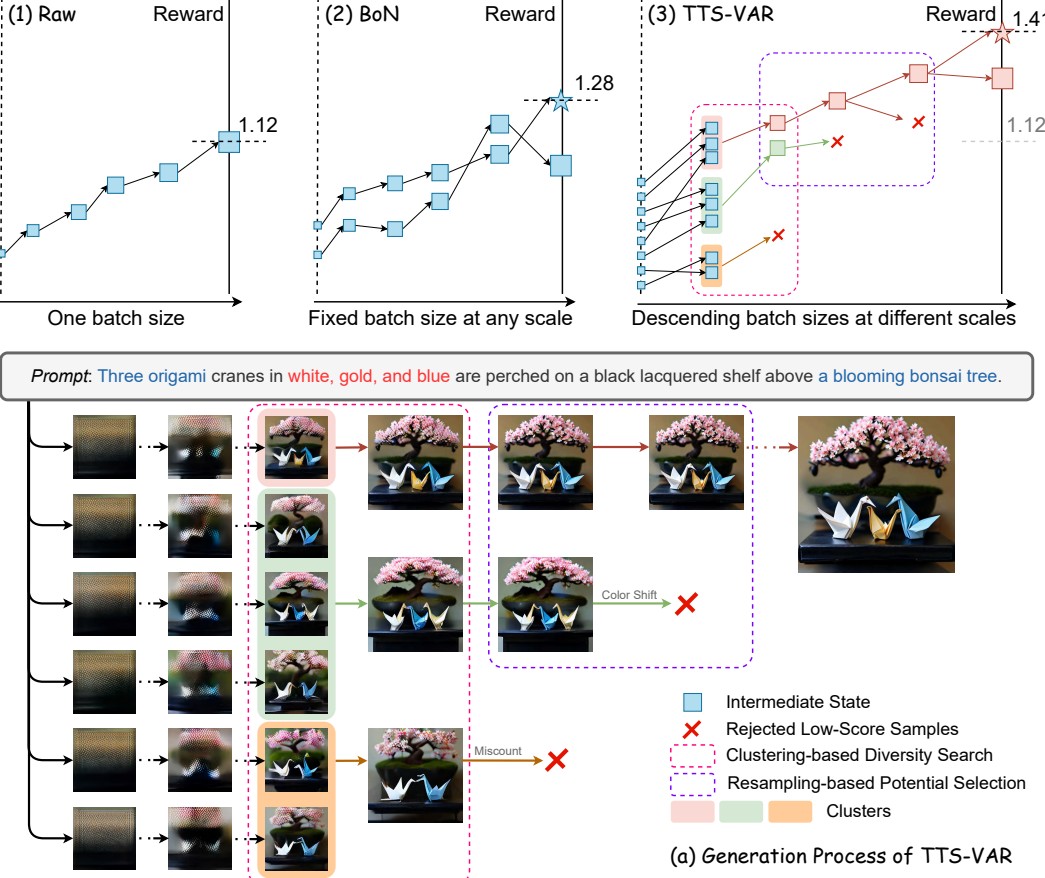

Figure 1: **TTS-VAR** generates several samples concurrently like Best-of-N (BoN). In *TTS-VAR*, we adopt an adaptive descending batch size schedule to make the most of AR efficiency, with feature clustering at early scales to ensure diversity, and resampling according to potentials at late scales for more valuable samples. (1-3) are overviews showing the difference between raw inference, BoN, and *TTS-VAR*. (a) is a detailed example of the generation process of our method.

Meanwhile, following the success of test-time scaling in LLMs [15–19] , researchers have begun exploring this methodology to image generative models for better results. In auto-regressive models, previous works typically formulate image generation as an image-level [20] or token-level [21] multi-stage process, treating the sequence of stages as the Chain-of-Thought (CoT). However, these methods require additional training to achieve effective scaling. Alternatively, diffusion-based approaches [22–26] regard scaling as a path searching problem, which scores different intermediate states and selects the most promising noise to denoise for higher-quality images. There are two main strategies to achieve great improvement. One [22, 23] introduces additional denoising steps to obtain the clean latents for image decoding and select intermediate latent states based on the final decoded results. The other [24], instead, directly scores decoded images from intermediate noisy latents to guide selection by reward functions [27].

Inspired by this perspective, we explore whether VAR models can also benefit from path searching. However, directly adopting the two strategies from diffusion models is non-trivial. For the former, the extra inference steps incur prohibitively high computation for VAR models with exponential complexity growth. They also disrupt the KV Cache mechanism [28, 29], which is crucial for retaining efficiency in AR inference. The latter also fails to reach the expectation. We observe that the rewards of images at early scales struggle to accurately represent the quality of final images, leading to incorrectly ruling out certain early-scale tokens which could be promising for later-scale generation, as demonstrated in Sec. 5.4. We attribute this to the difference between VAR and diffusion. In VAR, unlike diffusion process that can refine generated noise through iterative denoising, all tokens remain fixed once generated. Each token not only contributes to decoding the final image but also directly affects all subsequent token generation, resulting in much lower tolerance for poor early-stage tokens.

In this paper, we introduce the first Test-Time Scaling framework for VAR, abbreviated as *TTS-VAR*. Different from simple selection according to reward functions, we design scaling strategies aligned to the causal coarse-to-fine generation process of VAR. Firstly, noticing the progressively increased consumption of FLOPs and RAM in VAR, we implement our framework under an adaptive descending batch size schedule, reduced from larger batch sizes at coarse scales to smaller ones in fine scales. This promotes the expression of more possibilities with little additional consumption. Secondly, though early scales are hard to evaluate by reward functions, we observe that structural information, which has a great impact on image contents, can be captured since early scales, as shown in Fig. 1 (b) and Sec. 5.5. This motivates splitting the generation process into two key components: clustering-based diversity search for early scales and resampling-based potential selection for late scales. At early scales, while results of intermediate states can hardly be estimated, we aim to keep the diversity as batch size decreases, thus enabling later selections on samples with higher potential. We employ clustering on semantic features extracted by pre-trained extractors like DINOv2 [30], and pick from each category for dissimilar samples to ensure sampling diversity. At late scales, while scores of intermediate images share high consistency with those of final images, we calculate potential scores to directly resample preferred samples. The potential scores are specifically defined reward functions based on the generation history of all scales, instead of only the current one.

To summarize, we propose *TTS-VAR*, the first general test-time scaling framework for VAR models. By integrating clustering-based diversity search and resampling-based potential selection tailored to VAR's causal generation process, *TTS-VAR* consistently delivers stable performance improvements. We conduct comprehensive experiments and analysis on Infinity [14], a scaled-up text-to-image VAR model, revealing why resampling methods exhibit scale-dependent limitations and demonstrating the benefits of structural feature clustering. Notably, *TTS-VAR* significantly improves the GenEval score from 0.69 to 0.75, along with consistent improvements in other metrics.

## 2 Related Works

### 2.1 Test-time Scaling in Diffusion Models

Diffusion models [6, 31–34] create high-resolution images by denoising a Gaussian distribution into an image distribution. Initial research efforts [35, 5, 36] concentrated on scaling up the number of denoising steps to enhance the quality. However, it has been observed that as the number of inference steps increased, performance plateaued, and sampling additional steps is ineffective. Consequently, early research [37–39] mainly aims to reduce inference steps while maintaining image quality.

Ma et al. [22] address the scaling issue in diffusion models as a path searching problem, achieving significant improvements by applying several search strategies within the latent space, with reward functions serving as verifiers. Building on this problem definition, subsequent studies [23, 24] investigate the effectiveness of various search strategies and methods to accurately verify intermediate states for choice. Oshima et al. [23], for instance, employ few-step sampling instead of one-step sampling for denoised images that are clearer and more suitable for verification.

### 2.2 Test-time Scaling in Autoregressive Models

In autoregressive Large Language Models [9, 7, 8], test-time scaling is a widely employed technique to enhance performance. Since Wei et al. [15] proposed Chain-of-Thought and enabled LLMs to benefit from a structured thinking process, various studies [16–19] have explored tree search, graph search, and other methodologies to improve outcomes further. All these strategies leverage the reasoning capabilities of models and utilize the properties inherent in natural language for scaling.

However, within autoregressive image generative models [10, 13, 11, 40], characterized by a deterministic process with a steady token length, it is unnatural to directly increase the image token sequence as "thinking". Instead, Guo et al. [20] conceptualize generation CoT as an image-level problem. By employing a unified understanding and generation model [41] that first generates and then evaluates, it self-corrects results to align with expectations. Nevertheless, this approach relies solely on evaluating results, neglecting the generation process itself. Jiang et al. [21], instead, propose splitting the task into semantic-level and token-level phases, enabling a multi-stage generation as the thinking process. However, this method necessitates additional reinforcement learning for fine-tuning.

# 3 Preliminary: Visual Auto-Regressive Modeling

Unlike traditional next-token prediction auto-regressive models such as LLama-Gen [10], Visual Auto-Regressive Modeling (VAR) [13] tokenizes an input image $I$ into a feature map $F \in \mathbb{R}^{h \times w \times d}$. With quantizer $Q$, it quantizes the feature map $F$ into a sequence of multi-scale discrete residual feature maps [42] $\{r_k\}_{k=1}^{K}$, where $K$ represents the number of residual features across varying resolutions. For each residual feature map $r_k$, the resolution is $h_k \times w_k$, which progressively increases from $k = 1$ to $k = K$. Specifically, when $k = 1$, $h_k = w_k = 1$, and when $k = K$, $h_k = h, w_k = w$. From the sequence of residual features, at each scale $k$, a gradually refined feature map $f_k$ can be computed as:

$$f_k = \sum_{i=1}^{k} \text{up}(r_k, (h, w)), \tag{1}$$

where $\text{up}(\cdot, \cdot)$ denotes upsampling the single-scale feature map to the target resolution, and $f_k$ is the aggregated sum of features $\{r_k\}_{k=1}^{K}$. During inference, the downsampled accumulated feature map $\tilde{f}_k = \text{down}(f_k, (h_{k+1}, w_{k+1}))$ is appended as initial tokens for the prediction of the next scale and a scale-wise causal mask is employed to facilitate local bi-directional information modeling. The transformer is trained to predict the next-scale residual feature map. In Infinity [14], a VAR-based model scaled up for text-to-image generation, the quantizer $Q$ is advanced from VQ [43] to BSQ [44]. Additionally, a Flan-T5 [45] text encoder $\Psi$ is harnessed for prompt embeddings. With text prompt $c$ as the condition, the overall likelihood is:

$$p(r_1, r_2, \ldots, r_K) = \prod_{k=1}^{K} (r_k | r_1, r_2, \ldots, r_{k-1}; \Psi(c)). \tag{2}$$

# 4 Method

In *TTS-VAR*, we conceptualize the generation of high-quality and human-preferred images as a path searching problem, and identify two primary subproblems: (i) how to search for more possibilities, and (ii) how to select intermediate states for superior final results. Besides applying an adaptive batch size schedule as illustrated in Sec. 4.1 to enlarge the search scope, we introduce clustering-based diversity search in Sec. 4.2 and resampling-based potential selection in Sec. 4.3 to solve these problems. The complete method is illustrated in Fig. 1 (c).

## 4.1 Adaptive Batch Sampling

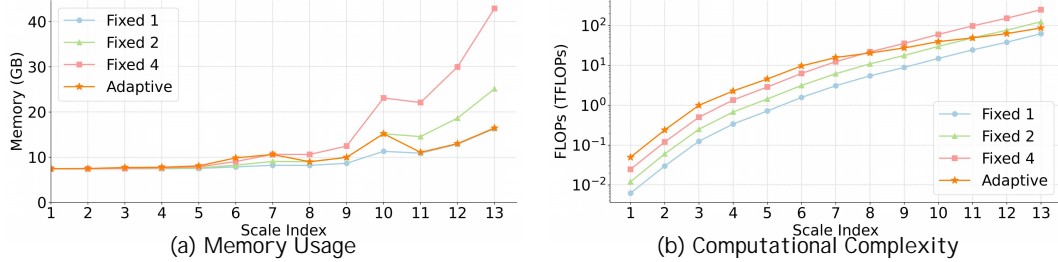

(a) Memory Usage          (b) Computational Complexity

Figure 2: **Different Batch Size Schedules.** We visualize the memory usage in (a) and computation complexity in (b) for 13 scales during the generation of Infinity, with fixed batch size 1 and adaptive batch size. Specifically, the adaptive batch size here is [8,8,6,6,6,4,2,2,2,1,1,1,1]. This batch size schedule enables more possibilities with little additional consumption.

Influenced by the causal attention mechanism, in VAR models, both RAM memory consumption and computational expense increase along with the length of the token sequence. As illustrated by the blue lines in Fig. 2 (a) and (b), during the inference process of early scales, memory requirements and computational costs are minimal. However, at later scales, when the pre-existing sequence extends significantly and the current prediction scale encompasses a large number of tokens, the resource consumption becomes substantial.

Therefore, we implement adaptive batch sizes during inference, capitalizing on the efficiency of lower consumption at early scales. This descending adaptive batch size schedule $\{b_0, b_1, \ldots, b_K\}$, where

$K$ represents the number of scales, generates more samples in earlier stages and fewer samples in the later stages. For a typical VAR model encompassing 13 scales, the batch size schedule is {*8N, 8N, 6N, 6N, 6N, 4N, 2N, 2N, 2N, 1N, 1N, 1N, 1N*} unless otherwise specified. At early scales, clustering in Sec. 4.2 filters several categories. At late scales, resampling in Sec. 4.3 chooses superior states. As demonstrated in Fig. 2, while the increased number of batches amplifies memory and computation costs at early scales, these additions are relatively minor in the overall expenditure.

## 4.2 Clustering-Based Diversity Search

As sequence length increases, maintaining a large batch size becomes cost-prohibitive, necessitating a filtering method for the desired samples. A straightforward approach involves calculating the reward function for intermediate results and selecting those with higher scores. However, our findings in Sec. 5.4 indicate that rating models struggle to evaluate early intermediate images for reward scores consistent with the final images, as also observed by Guo et al. [20]. To avoid erroneous elimination of certain initial samples that may hold potential for later-scale generation, we explore ways to keep the diversity of samples.

We observe that during the generation process, unlike the details appearing in late scales, the structural information, which significantly influences the quality of final images, is conveyed from the early phases. We analyze this phenomenon in Sec. 5.5 and find extractors like DINOv2 [30] can effectively capture features strongly connected with structures. Given this, we create clusters based on the semantic features to filter samples from each category and ensure structural diversity, thereby maximally enhancing possibilities for valuable results.

Specifically, from the current batch size $b_i$, we need to select $b_{i+1}$ samples as next states. Firstly, for $b_i$ intermediate images $\{I_j\}_{j=1}^{b_i}$, each image is embedded in a high-dimensional semantic embedding space via a feature extractor $F$, creating a set of embeddings $S = \bigcup_j F(I_j)$. Subsequently, we apply the K-Means++ [46] algorithm to cluster these embeddings into $b_{i+1}$ cluster centers and select samples with the shortest L2 distance to cluster centers as new batches.

To extract structural information for diversity, we primarily employ the self-supervised DINOv2 [30] as the extractor, generating the feature map $s \in \mathbb{R}^{(h' \times w') \times d}$. To obtain one-dimensional features for clustering, we apply PCA reduction on feature patches for $s' \in \mathbb{R}^{(h' \times w')}$. We also consider pooling the second dimension for $s' \in \mathbb{R}^d$, and supervised features from InceptionV3 [47] without the final fully connected layer. We discuss these choices in Sec. 5.5.

## 4.3 Resampling-Based Potential Selection

In contrast to early-stage diversity preservation through clustering, when intermediate images show high consistency with final results, reward functions can directly guide the generation toward higher quality and alignment with human preferences at late scales. Typically, a reward function $r_\phi(x)$ is derived from a reward model $\phi$, which includes specially trained rating models and vision-language models. For scores conditioned on a text prompt $c$, the reward function can be expressed as $r_\phi(x, c)$. In the context of a generative model based on the generation distribution $p_\theta(x)$, we aim to steer the distribution to align with reward preferences [48, 24], as follows:

$$p_{\theta'}(x) = \frac{1}{Z} p_\theta(x) \exp(\lambda \cdot r_\phi(x, c)) \tag{3}$$

where $p_{\theta'}(x)$ is the target distribution, $Z$ is a normalization constant, and $\lambda$ is a hyperparameter to control the temperature in selection.

To obtain high-quality samples, we evaluate the reward score for each intermediate state at current scale $k$ and replace them with ones sampled from a multinomial distribution based on the potential score $P_k$. Considering that the generation of VAR is a path with historical states and merely rating the image $x_k = \mathcal{D}(f_k)$, decoded by the image decoder $\mathcal{D}$ from the accumulated feature $f_k$, may not adequately reflect the potential of final results, we therefore contemplate several potential scores.

**Potential Score $P_k$.** We denote $P_k(x_0, x_1, \ldots, x_k)$ as the potential score of a sample at scale $k$, with $x_0, x_1, \ldots, x_k$ representing the generation history of this sample.

- $P_k(x_0, x_1, \ldots, x_k) = \exp(\lambda \cdot r_\phi(x_k, c))$: This directly utilizes the reward score as the potential score, referred to as Value. It is also known as importance sampling (IS) [49, 50].

| Methods | # Params | GenEval↑ | | | |
|---|---|---|---|---|---|
| | | Two Obj. | Counting | Color Attri. | **Overall** |
| *Diffusion Models* | | | | | |
| SDXL [2] | 2.6B | 0.74 | 0.39 | 0.23 | 0.55 |
| +FK ($N = 8$) [24] | 2.6B | - | - | - | 0.65 |
| PixArt-Alpha [1] | 0.6B | 0.50 | 0.44 | 0.07 | 0.48 |
| DALL-E 3 [3] | - | 0.87 | 0.47 | 0.45 | 0.67 |
| FLUX [4] | 12B | 0.81 | 0.74 | 0.45 | 0.66 |
| SD3 [60] | 8B | 0.94 | 0.72 | 0.60 | 0.74 |
| *AR Models* | | | | | |
| LlamaGen [10] | 0.8B | 0.34 | 0.21 | 0.04 | 0.32 |
| Chameleon [61] | 7B | - | - | - | 0.39 |
| Show-o [41] | 1.3B | 0.52 | 0.49 | 0.28 | 0.53 |
| +PARM ($N = 20$) [20] | 1.3B | 0.77 | 0.68 | 0.45 | 0.67 |
| Emu3 [11] | 8.5B | 0.71 | 0.34 | 0.21 | 0.54 |
| Infinity | 2B | 0.8351 | 0.5923 | 0.6150 | 0.6946 |
| Infinity+IS ($N = 8$) | 2B | 0.8969 | 0.6220 | 0.6550 | 0.7181 |
| Infinity+BoN ($N = 8$) | 2B | 0.9201 | 0.6756 | 0.6700 | 0.7364 |
| **Infinity+Ours** ($N = 2$) | 2B | 0.9278 | 0.7113 | 0.6775 | 0.7403 |
| **Infinity+Ours** ($N = 8$) | 2B | **0.9501** | **0.7411** | **0.6800** | **0.7530** |
| Infinity | 8B | 0.8866 | 0.7292 | 0.6750 | 0.7646 |
| Infinity+BoN ($N = 4$) | 8B | 0.9175 | 0.7946 | 0.7225 | 0.7995 |
| **Infinity+Ours** ($N = 2$) | 8B | **0.9330** | 0.7798 | 0.7050 | 0.7985 |
| **Infinity+Ours** ($N = 4$) | 8B | 0.9304 | **0.8036** | **0.7600** | **0.8188** |

Table 1: **Quantitative evaluation on GenEval**.

Figure 3: **Score Curves over Sample Number $N$**.

- $P_k(x_0, x_1, \ldots, x_k) = \exp(\lambda \cdot (r_\phi(x_k, c) - r_\phi(x_{k-1}, c)))$: This computes the difference between two consecutive scales as the potential score, termed DIFF.

- $P_k(x_0, x_1, \ldots, x_k) = \exp(\lambda \cdot \max_{i=0}^{k}\{r_\phi(x_i, c)\})$: This selects the highest score in the generation path as the current potential score, designated MAX.

- $P_k(x_0, x_1, \ldots, x_k) = \exp(\lambda \cdot \sum_{i=0}^{k} r_\phi(x_i, c))$: This accumulates all scores from the history to determine the current potential score, labeled SUM.

Different potential scores benefit distinct attributes of the generation history. For instance, DIFF favors samples with higher growth rates, while MAX favors those with higher ceilings. In our setting, VALUE performs well. We will explore these choices in Sec. 5.4.

## 5 Experiments

In this section, we demonstrate the effectiveness of our *TTS-VAR* on the powerful VAR model Infinity [14] with resampling temperature $\lambda = 10$. We present the comparisons in Sec. 5.1 and Sec. 5.2, and precisely analyze design details in Sec. 5.4 and Sec. 5.5. Following previous work [24, 22, 51], we utilize ImageReward [52] as the reward function for guidance. We evaluate results using the main metric GenEval [53], and T2I-CompBench [54], with relevant indicators ImageReward [52], HPSv2.1 [55, 56], Aesthetic V2.5 [57], and CLIP-Score [58, 59], based on prompts offered by GenEval.

### 5.1 Overall Performance

As shown in Table 1, our proposed method demonstrates significant improvements over existing state-of-the-art models and conventional test-time scaling strategies (Importance Sampling and Best-of-N). With a model size of 2B parameters, *TTS-VAR* achieves an overall GenEval score of 0.7530 at $N = 8$, surpassing the record 0.74 of Stable Diffusion 3 (8B parameters) while utilizing 60% fewer parameters. Our framework also exhibits substantial gains across single items, like two objects. Notably, even with minimal computational overhead ($N = 2$), our approach attains the competitive performance of score 0.7403, outperforming Best-of-N ($N = 8$) with only 25% sample number.

From the perspective of different $N$, *TTS-VAR* maintains consistent performance gains across varying sample sizes in both GenEval and ImageReward metrics, as illustrated in Figure 3. While Best-of-N

| Model | Avg. | Color | Shape | Texture | 2D Spatial | 3D Spatial | Numeracy | Non-spatial | Complex |
|---|---|---|---|---|---|---|---|---|---|
| | | B-VQA | B-VQA | B-VQA | UniDet | UniDet | UniDet | S-CoT | S-CoT |
| Stable v2 [31] | 0.4839 | 0.5065 | 0.4221 | 0.4922 | 0.1342 | 0.3230 | 0.4582 | 0.7567 | 0.7783 |
| Stable XL [2] | 0.5255 | 0.5879 | 0.4687 | 0.5299 | 0.2133 | 0.3566 | 0.4988 | 0.7673 | 0.7817 |
| Pixart-$\alpha$-ft [1] | 0.5583 | 0.6690 | 0.4927 | 0.6477 | 0.2064 | 0.3901 | 0.5032 | 0.7747 | 0.7823 |
| DALLE· 3 [3] | 0.6168 | 0.7785 | **0.6205** | 0.7036 | **0.2865** | 0.3744 | 0.5926 | 0.7853 | 0.7927 |
| FLUX.1 [4] | 0.6087 | 0.7407 | 0.5718 | 0.6922 | 0.2863 | 0.3866 | 0.6185 | 0.7809 | 0.7927 |
| Infinity2B [14] | 0.5688 | 0.7421 | 0.4557 | 0.6034 | 0.2279 | 0.4023 | 0.5479 | 0.7820 | 0.7890 |
| Infinity2B+IS ($N = 8$) | 0.5965 | 0.7746 | 0.5078 | 0.6501 | 0.2462 | 0.4194 | 0.6002 | 0.7803 | 0.7937 |
| Infinity2B+BoN ($N = 8$) | 0.6115 | 0.7950 | 0.5439 | 0.6886 | 0.2545 | 0.4205 | 0.6090 | 0.7870 | 0.7937 |
| **Infinity2B+Ours** ($N = 2$) | 0.6151 | 0.7887 | 0.5578 | 0.6858 | 0.2697 | 0.4286 | 0.6112 | 0.7853 | 0.7936 |
| **Infinity2B+Ours** ($N = 8$) | **0.6230** | **0.8073** | 0.5914 | **0.7121** | 0.2644 | **0.4302** | **0.6340** | **0.7880** | **0.7963** |

Table 2: **Quantitative evaluation on T2I-CompBench**.

sampling benefits more from larger N, its performance remains distinctly inferior to ours, even failing to match our results at $N = 2$ with $N = 8$. Detailed evaluations across different $N$ values are provided in the appendix for comprehensive analysis.

To better demonstrate the generalizability of our framework across different models, we also conducted comparative experiments on the Infinity8B model. The results show that even on this stronger 8B model, our approach effectively boosts performance, improving accuracy from 0.76 to 0.82—an increase of about 7Moreover, it surpasses the gains achieved by methods such as IS and BoN under the same settings. This indicates that our method can be directly applied to other VAR-based architectures.

We further evaluated performance on T2I-CompBench [54]. As presented in Table 2, *TTS-VAR* exhibits a remarkable improvement across every indicator compared to the base model. Consistent with the consequences on GenEval, our method achieves superior results at $N = 2$ compared to Best-of-N at $N = 8$, and secures the highest scores on most individual items and the overall average.

## 5.2 Qualitative Comparison

We present images generated by different methods here for reference, to further explain the improvement in image quality and text alignment. As displayed in Fig. 4, our method correctly generated objects with the required number, for example, the "kayaks" in the first case and the "birds" in the second case, which is hard for base models and other scaling strategies to achieve. In the complex scenes with several pairs of numbers and colors, *TTS-VAR* ensures the color attribute like the "golden apples" in the fourth case, avoiding the problem of attribute forgetting.

## 5.3 User Study

To evaluate from the view of practical usage and gather user feedback, we selected a small set of 15 image groups and compiled them into a questionnaire on Google Forms. For each group, the images are randomly shuffled, and participants were asked to select the best generated image based on three criteria: Image Quality, Image Rationality, and Consistency with the Prompt. In total, we collected 21 completed questionnaires, resulting in 315 sets of survey data for each indicator. The percentages of votes received by each approach are summarized in Table 3 below.

| Metric | Baseline | IS | BoN | Ours |
|---|---|---|---|---|
| Image Quality | 13.3% | 7.9% | 13.3% | **65.4%** |
| Image Rationality | 13.7% | 8.6% | 8.6% | **69.2%** |
| Consistency with the Prompt | 1.3% | 1.9% | 2.5% | **94.3%** |

Table 3: **User Study**.

## 5.4 Resampling for Superior Samples

**Analysis**. In path searching, resampling intermediate states with high potentials is a straightforward yet effective approach for superior results with minimal consumption. However, in VAR, this may not

SD3 Infinity Infinity-IS Infinity-BoN Infinity-TTS-VAR

*Prompt:* Two pink flamingos standing on a wooden dock, with three turquoise kayaks floating on the calm lake behind them.

*Prompt:* An oil painting, where a green vintage car, a blue scooter on the left of it and a black bicycle on the right of it, are parked on the road, with two birds in the sky

*Prompt:* A blooming garden path shows two orange marigolds leaning toward one yellow sunflower, all surrounded by lush green leaves, with two butterflies hovering above them in a V-formation.

*Prompt:* An elegant dining table with a dark wooden surface holds two ivory candles burning gently, a crystal vase with three red roses, and a silver tray holding two golden apples.

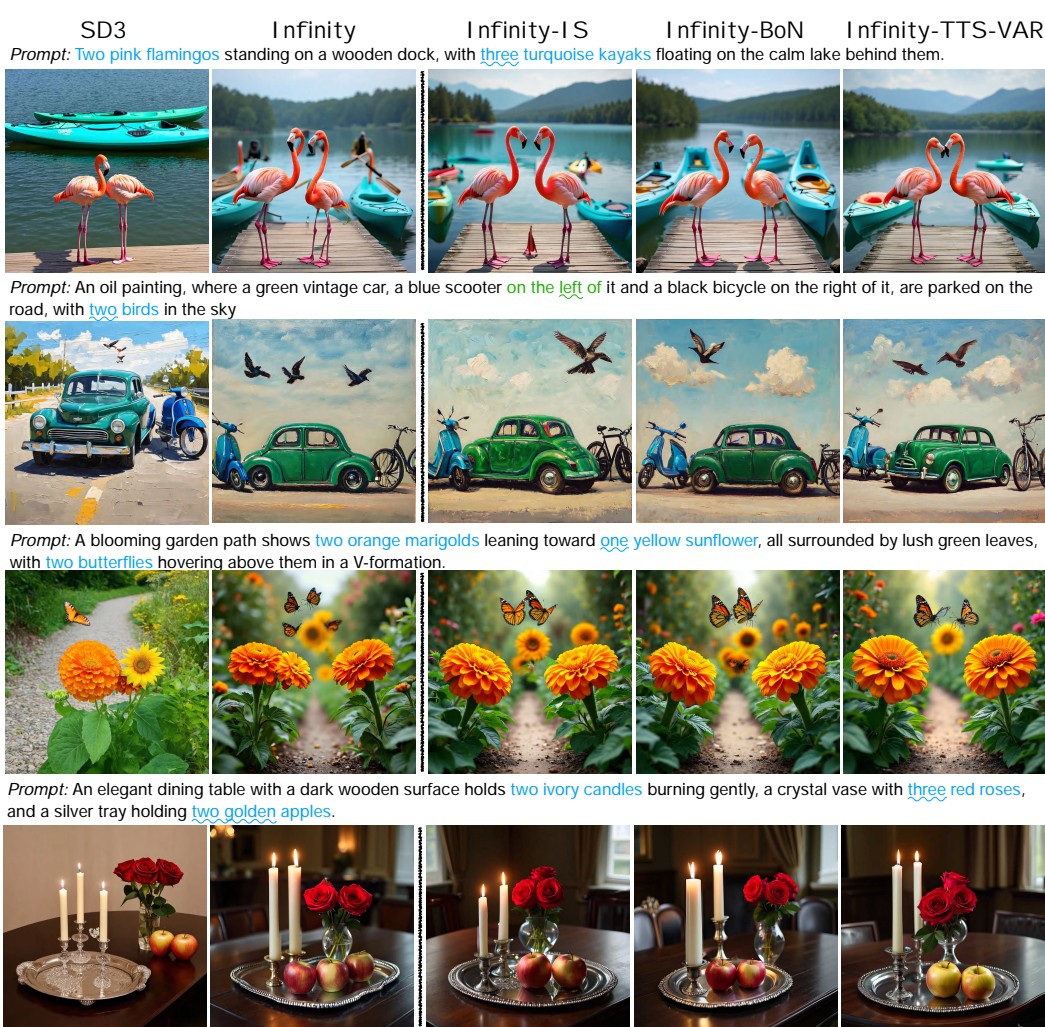

Figure 4: **Qualitative Comparison.** Each line shows results generated by Stable Diffusion 3 (SD3) [60], Infinity, and Infinity with test-time scaling strategies, with objects marked blue and relationships marked green.

be advantageous and can even be detrimental at certain scales. In Fig. 5 (a), we illustrate the score distinctions between solely employing Best-of-N ($N = 2, 4$) and concurrently applying potential (VALUE) resampling at specific scales. Although Best-of-N selection ensures comparatively high results, it is evident that resampling at early scales (e.g., scale 3) leads to a noticeable decline in the final results. Instead, resampling at later scales yields a certain degree of improvement.

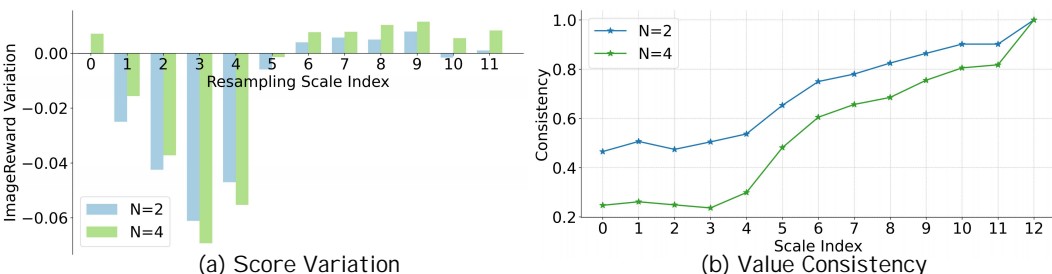

(a) Score Variation      (b) Value Consistency

Figure 5: **Resample choices.** The left graph shows the variation in ImageReward Score when executing resampling-based potential selection at different scales (0-11). The right graph shows the consistency between scores of intermediate states and those of final results at each scale. It demonstrates that in VAR, not all scales are suitable for selection, and some may lead to degradation.

| $N$ | Resampling Scale | GenEval | ImageReward | HPS | CLIP | Aesthetic |
|---|---|---|---|---|---|---|
| 1 | - | 0.6946 | 1.132 | 0.3042 | 0.3366 | 0.5811 |
| 2 | [6, 9] | **0.7133** | 1.2572 | 0.3066 | 0.3379 | 0.5801 |
| 2 | [6, 8, 10] | 0.7130 | **1.2591** | 0.3066 | **0.3381** | 0.5809 |
| 2 | [6, 7, 8, 9, 10, 11] | 0.7114 | 1.2497 | **0.3067** | 0.3378 | **0.5810** |
| 4 | [6, 9] | **0.7276** | 1.3534 | 0.3082 | **0.3398** | 0.5817 |
| 4 | [6, 8, 10] | 0.7247 | **1.3592** | **0.3085** | 0.3397 | 0.5822 |
| 4 | [6, 7, 8, 9, 10, 11] | 0.7210 | 1.3558 | 0.3083 | 0.3398 | **0.5830** |

Table 4: **Resampling Scale Difference.** This table shows results with different resampling scales.

We analyze this phenomenon from the perspective of consistency between intermediate states and final images, as shown in Fig. 5 (b). We compute potential scores at each scale and select the one with the highest potential accordingly. We then assess whether the selected best intermediate state aligns with the best final image (the optimal state at scale 12), resulting in a sequence of scores between 0 and 1, termed **consistency**. Low consistency of early scales in this evolving curve indicates that those scores rarely accurately reflect the quality of final results. The scores become valuable from a certain late scale, like scale 6, with comparatively high consistency. This explains why resampling efficacy varies and should be applied selectively on later scales.

**Resampling Scales**. Based on the aforementioned observation, we further investigate whether increasing the resampling frequency at late scales is beneficial. As illustrated in Table 4, compared with raw inference, resampling greatly enhances the results. However, increasing the frequency has a negligible impact. For instance, there is a modest improvement in ImageReward and HPS for scales [6,8,10] compared to scales [6,9], but at the cost of Geneval. Considering that executing resampling incurs additional computational expenses related to image decoding and score calculation, we opt to resample only at scales 6 and 9.

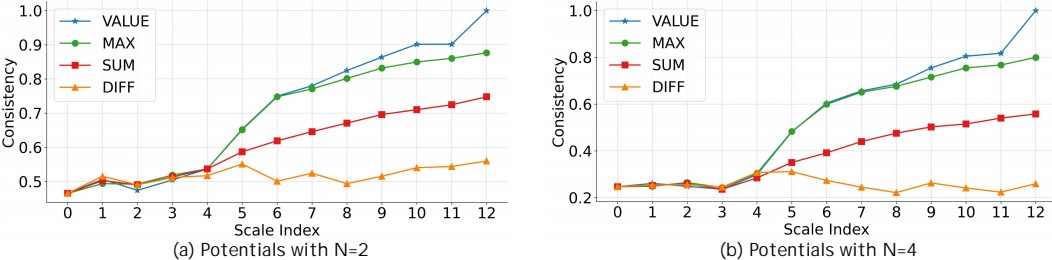

(a) Potentials with N=2      (b) Potentials with N=4

Figure 6: **Consistency of Different Potentials.** We visualize the consistency between different pairs of potential scores and final results. Accordingly, VALUE and MAX can better indicate the potentials.

**Potential Scores**. As mentioned in Sec. 4.3, we have developed distinct computational modes to explore those with higher potentials. Initially, we seek better options through a theoretical approach by visualizing the consistency. As exhibited, DIFF continuously yields low consistency levels and fails to predict the desired outcomes. SUM demonstrates a stable increment but with comparatively low values. In contrast, VALUE and MAX exhibit similar characteristics, maintaining relatively high scores since scale 6 and showing a steady increase.

Experiments in Table 5 with $N = 2, 4$ present coherent results. Among these potentials, DIFF lags in all indicators. Although SUM achieves some acceptable outcomes, the overall score remains low. As forecasted by consistency, VALUE and MAX achieve the highest scores in text-related metrics such as GenEval, ImageReward, and HPS, indicating a higher likelihood of selecting for superior final results. Considering that MAX requires score calculations at each scale and leads to an additional computational cost, we utilize VALUE as the potential score.

| $N$ | Potential | GenEval | ImageReward | HPS | CLIP | Aesthetic |
|---|---|---|---|---|---|---|
| 2 | VALUE | 0.7133 | **1.2572** | **0.3066** | **0.3379** | 0.5801 |
| 2 | MAX | **0.7150** | 1.2510 | 0.3065 | 0.3379 | **0.5803** |
| 2 | SUM | 0.7130 | 1.2364 | 0.3064 | 0.3379 | 0.5801 |
| 2 | DIFF | 0.7006 | 1.1725 | 0.3042 | 0.3365 | 0.5798 |
| 4 | VALUE | 0.7276 | **1.3534** | **0.3082** | **0.3398** | 0.5817 |
| 4 | MAX | **0.7285** | 1.3495 | 0.3082 | 0.3398 | 0.5815 |
| 4 | SUM | 0.7244 | 1.3326 | 0.3082 | 0.3394 | **0.5830** |
| 4 | DIFF | 0.7030 | 1.2412 | 0.3051 | 0.3378 | 0.5808 |

Table 5: **Potential Score Difference.** This table shows results with different potential scores.

## 5.5 Clustering for Structural Diversity

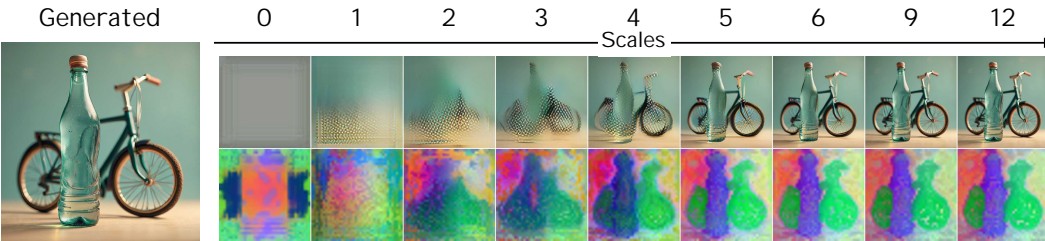

Figure 7: **Visualization of Generation Process.** The text prompt is "a photo of a bottle and a bicycle". The left is the generated image. The right is the generation process and visualized DINOv2 features from scales. It demonstrates that features in early scales can indicate the structural information.

**Analysis**. In reference to Sec. 5.4, resampling is not universally applicable to all scales. Nevertheless, in VAR, the effectiveness and low cost associated with early scales present an invaluable and unmissable opportunity to search for more samples, thereby unlocking greater potential for the final outcomes. We notice that, given the same prompt, the structure of images significantly influences the scores. Moreover, unlike details that emerge later, structural information can be captured from the early scales. The right side of Fig. 7 demonstrates that the generation process follows a structure-to-detail progression, with rough outlines becoming discernible from scale 2. When using DINOv2 to extract intermediate images and visualizing them through PCA, as seen in the bottom line, these features exhibit characteristics akin to the original images. Consequently, we leverage structural information and conduct clustering-based diversity search to sample dissimilar structures, thus scaling for more possibilities, especially when resampling may not suffice.

| $N$ | Clustering Scale | GenEval | ImageReward |
|---|---|---|---|
| 2 | - | 0.7087 | 1.2545 |
| 2 | [2] | 0.7089 | 1.2513 |
| 2 | [5] | 0.7099 | 1.2558 |
| 2 | [2, 5] | **0.7184** | **1.2682** |
| 4 | - | 0.7244 | 1.3471 |
| 4 | [2] | 0.7300 | 1.3502 |
| 4 | [5] | 0.7293 | 1.3558 |
| 4 | [2, 5] | **0.7337** | **1.3610** |

Table 6: **Clustering Scale Difference.** This table shows results with and without clustering at certain scales.

| $N$ | Extractor | GenEval | ImageReward |
|---|---|---|---|
| 2 | PCA | **0.7184** | 1.2682 |
| 2 | Pool | 0.7127 | 1.2720 |
| 2 | Inception | 0.7073 | **1.2727** |
| 4 | PCA | **0.7337** | 1.3610 |
| 4 | Pool | 0.7296 | 1.3629 |
| 4 | Inception | 0.7207 | **1.3664** |

Table 7: **Extractor Difference.** This table shows results when adopting different feature extraction methods. PCA and Pool here are both transformed from 2-dimensional features extracted by DINOv2.

**Clustering Scales**. According to Fig. 7, features at scale 2 display a coarse structure, while those at scale 5 reveal a refined structure similar to final outcomes. Therefore, we specifically apply clustering on these scales. The results of $N = 2, 4$, with and without clustering, are presented in Table 6. The first lines of each block denote the results without clustering (Best-of-N), with subsequent lines showing outcomes under different clustering scales. Evidently, each clustering increases the likelihood of yielding better results, and there is an obvious growth when employing both scales.

**Extractor**. We tested various extractors for clustering features outlined in Table 7. Both PCA and Pool are transformations of features extracted by DINOv2 [30], as detailed in Sec. 4.2. While supervised InceptionV3 [47] features perform optimally in ImageReward, they notably underperform in GenEval. PCA delivers superior results on average and is employed. We attribute this to that the patch-level features from PCA align more closely with the observed structural traits.

## 6 Conclusion

In this work, we introduce the first general test-time scaling framework for VAR models. Through analysis on different scales, we demonstrate that *TTS-VAR*, which incorporates adaptive batch sampling, clustering-based diversity search, and resampling-based potential selection, aligns with distinct stages of VAR generation process. This dual-strategy approach enhances final result quality with minimal additional computational cost while maintaining algorithmic efficiency. We notice the limitation and potential societal impact on privacy and copyright, and discuss these in the appendix.

## Acknowledgements

This work is supported by the National Nature Science Foundation of China (No. 62402406).

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

# Appendix

## A Algorithm of *TTS-VAR*

We describe the algorithm of *TTS-VAR* in Alg. 1. Following the generation process of VAR [13] (Infinity [14]), *TTS-VAR* first predicts the residual tokens at the current scale and adds them to the accumulated feature maps. At scales that require clustering, *TTS-VAR* uses the extractor to gather features from $b_i$ intermediate images decoded from the feature maps. It then clusters the samples based on these features and selects $b_{i+1}$ ones as the next batch. At scales that require resampling, *TTS-VAR* employs the potential function to calculate scores for each image and samples $b_{i+1}$ indexes from the multinomial distribution for superior intermediate states.

---

**Algorithm 1** *TTS-VAR*

---

**Require:** Scales $\mathcal{S} = \{s_1, s_2, ..., s_K\}$, Descending batch sizes $\mathcal{B} = \{b_1, b_2, ..., b_K\}$, Clustering scales set $S_c$, Resampling scales set $S_r$, Generative model $\theta$, Reward model $r_\phi$ Extractor $F$, Potential Score function $P$, Text prompt $c$.

1: Initialize accumulated feature map $f_0$ with zeros.
2: **for** $i \in \{1, 2, ..., K\}$ **do**                                         ▷ Iterate through scales
3:     $r_i \leftarrow \text{Generate}(\theta, b_i, s_i, f_{i-1}, c)$
4:     $f_i \leftarrow f_{i-1} + r_i$
5:     **if** $s_i \in S_c$ **then**                                              ▷ Clustering phase
6:         $x \leftarrow \text{Decode}(f_i)$
7:         $feat \leftarrow F(x)$
8:         $index \leftarrow \text{KMeans++}(feat, b_{i+1})$
9:         $f_i \leftarrow f_i[index]$
10:    **else if** $scale \in S_r$ **then**                                       ▷ Resampling phase
11:        $x \leftarrow \text{Decode}(f_i)$
12:        $rw \leftarrow r_\phi(x)$
13:        $p \leftarrow P(rw)$
14:        $index \leftarrow \text{Multinomial}(p, b_{i+1})$
15:        $f_i \leftarrow f_i[index]$
16:    **end if**
17: **end for**
    **return** Final generated images $\text{Decode}(f_K)$

---

## B Hyperparameters Settings

In K-Means, we set the next batch size $b_{i+1}$ in the adaptive descending batch sizes as the number of centers, to find $b_{i+1}$ different structure categories. For PCA, we only select the first major component, as we utilize it for dimension reduction only. In all benchmarks, we use seeds from *0* to *3* for different results. For experiments requiring multi-GPU, like $N = 8$, we set the seed from *100\*process_index* to *100\*process_index+3* for each device. We use adaptive batch size [8*N*,8*N*,6*N*,6*N*,6*N*,4*N*,2*N*,2*N*,2*N*,1*N*,1*N*,1*N*,1*N*] for Infinity2B with **1024 pixel** and [8*N*,8*N*,6*N*,6*N*,6*N*,4*N*,2*N*,2*N*,1*N*,1*N*] for Infinity8B with **512 pixel**.

For other parameters related to Infinity, we keep them the same as the original public settings.

## C Detailed Main Results

We present detailed Infinity2B results of variant curves in Table 6. As evident, *TTS-VAR* demonstrates clear advantages across all indicators [53, 52, 56, 57, 59] compared to the baselines. In Table 7, we list each item of the GenEval [53] metric. Generally, our method significantly improves performance on handling two objects and counting tasks. We attribute this to the importance of structural accuracy in multi-character scenes, particularly when two objects and multiple identical objects (counting) are involved. For instance, when provided with a prompt for three objects, there is a possibility that the model may incorrectly generate a layout with four objects. Once this error occurs, following the

| $N$ | Strategy | GenEval | ImageReward | HPS | CLIP | Aesthetic |
|---|---|---|---|---|---|---|
| 1 | Raw Inference | 0.6946 | 1.1320 | 0.3042 | 0.3366 | 0.5811 |
| 1 | Ours | **0.7253** | **1.3226** | **0.3084** | **0.3395** | **0.5822** |
| 2 | Importance Sampling | 0.7022 | 1.1941 | 0.3051 | 0.3374 | 0.5807 |
| 2 | Best-of-N | 0.7087 | 1.2545 | 0.3069 | 0.3384 | 0.5813 |
| 2 | Ours | **0.7403** | **1.4136** | **0.3106** | **0.3411** | **0.5821** |
| 4 | Importance Sampling | 0.7116 | 1.2883 | 0.3067 | 0.3387 | 0.5815 |
| 4 | Best-of-N | 0.7244 | 1.3471 | 0.3083 | 0.3397 | 0.5820 |
| 4 | Ours | **0.7437** | **1.4605** | **0.3112** | **0.3414** | **0.5821** |
| 8 | Importance Sampling | 0.7181 | 1.3657 | 0.3085 | 0.3395 | 0.5810 |
| 8 | Best-of-N | 0.7364 | 1.4144 | 0.3103 | 0.3406 | **0.5820** |
| 8 | Ours | **0.7530** | **1.4995** | **0.3122** | **0.3420** | 0.5810 |

Table 6: **Scores over Different Strategies of Infinity2B.**

structure-to-detail generation process in VAR, it becomes challenging for subsequent scales to rectify. However, *TTS-VAR* facilitates structure diversity, thereby enabling the selection of a layout with the correct configuration and avoiding the irreversible wrong generation process for inferior samples.

| $N$ | Strategy | Overall | Single Obj. | Two Obj. | Counting | Colors | Position | Color Attri. |
|---|---|---|---|---|---|---|---|---|
| 1 | Raw Inference | 0.6946 | 0.9938 | 0.8351 | 0.5923 | **0.9293** | 0.2020 | 0.6150 |
| 1 | Ours | **0.7253** | 0.9938 | 0.9072 | 0.6518 | 0.9192 | **0.2096** | **0.6700** |
| 2 | Importance Sampling | 0.7022 | **0.9969** | 0.8497 | 0.6071 | 0.9268 | 0.1869 | 0.6475 |
| 2 | Best-of-N | 0.7087 | 0.9906 | 0.8789 | 0.6339 | 0.9242 | 0.1944 | 0.6300 |
| 2 | Ours | **0.7403** | 0.9936 | **0.9278** | **0.7113** | **0.9318** | **0.1995** | **0.6775** |
| 4 | Importance Sampling | 0.7116 | 0.9906 | 0.8840 | 0.6339 | **0.9318** | 0.1970 | 0.6325 |
| 4 | Best-of-N | 0.7244 | **1.0000** | 0.8969 | 0.6756 | 0.9242 | 0.1944 | 0.6550 |
| 4 | Ours | **0.7437** | 0.9906 | **0.9510** | **0.6994** | 0.9293 | **0.2045** | **0.6875** |
| 8 | Importance Sampling | 0.7181 | 0.9906 | 0.8969 | 0.6220 | 0.9318 | 0.2121 | 0.6550 |
| 8 | Best-of-N | 0.7364 | 0.9938 | 0.9201 | 0.6756 | **0.9444** | 0.2146 | 0.6700 |
| 8 | Ours | **0.7530** | **0.9969** | **0.9501** | **0.7411** | 0.9318 | **0.2172** | **0.6800** |

Table 7: **GenEval Details.** This table shows each item of the GenEval benchmark. "Object" is short for "Obj.", and "Attribute" is short for "Attri.".

In Table 8, we present the detailed different scores of Infinity8B. As shown, our method promises a stable increase in different models, including ones with superior performance.

| $N$ | Strategy | GenEval | ImageReward | HPS | CLIP | Aesthetic |
|---|---|---|---|---|---|---|
| 1 | Raw Inference | 0.7646 | 1.2095 | 0.3081 | 0.3388 | 0.5721 |
| 1 | Ours | **0.7931** | **1.3929** | **0.3115** | **0.3414** | **0.5727** |
| 2 | Importance Sampling | 0.7834 | 1.3091 | 0.3097 | 0.3399 | 0.5716 |
| 2 | Best-of-N | 0.7880 | 1.3435 | **0.3127** | **0.3423** | 0.5722 |
| 2 | Ours | **0.7985** | **1.4572** | 0.3106 | 0.3411 | **0.5730** |
| 4 | Importance Sampling | 0.7901 | 1.3786 | 0.3110 | 0.3409 | 0.5722 |
| 4 | Best-of-N | 0.7995 | 1.4092 | 0.3122 | 0.3397 | **0.5730** |
| 4 | Ours | **0.8189** | **1.5100** | **0.3139** | **0.3425** | 0.5727 |

Table 8: **Scores over Different Strategies of Infinity8B.**

# D    Results on DPG-Bench

We evaluate the performance of the baseline Infinity model, the Best-of-N (BoN) method, and our approach on the DPG-Bench, specifically for the case where $N = 4$. For each prompt, we generated four corresponding images for evaluation using seeds 0 through 3. The results are presented in Table 9.

# E    Performance over Computational Consumption

We here display the changing curves of GenEval, ImageReward, and HPSv2 over the increment of computation in Fig. 8, along with the increment of sample number $N$. As shown, our method *TTS-VAR* has higher computational efficiency and surpasses Importance Sampling and Best-of-N with less than half TFLOPs.

| Methods | Global | Relation | Overall |
|---|---|---|---|
| SDv2.1 | 77.67 | 80.72 | 68.09 |
| DALL-E 3 | 90.97 | 90.58 | 83.50 |
| SDXL | 83.27 | 86.76 | 74.65 |
| PixArt-Sigma | 86.89 | 86.59 | 80.54 |
| SD3 (d=24) | - | - | 84.08 |
| HART | - | - | 80.89 |
| Show-o | - | - | 67.48 |
| Emu3 | - | - | 81.60 |
| Infinity | 80.96 | 89.83 | 81.72 |
| Infinity2B+BoN ($N = 4$) | 88.02 | 87.33 | 82.52 |
| **Infinity2B+Ours** ($N = 4$) | 83.37 | 88.81 | 82.94 |

Table 9: Evaluation results on **DPG-Bench** with $N = 4$.

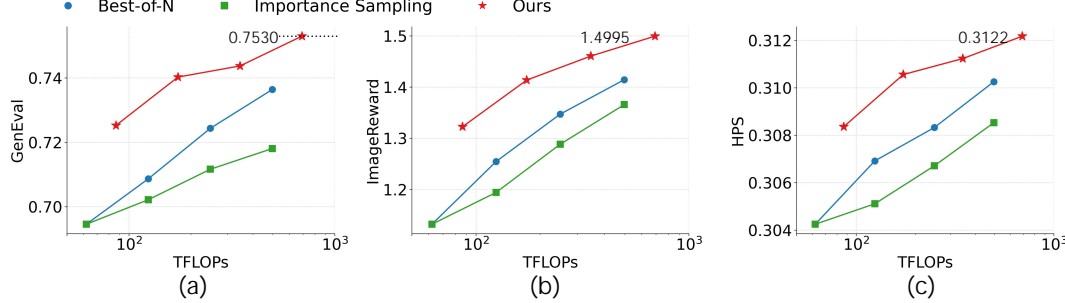

(a) (b) (c)

Figure 8: **Performance over Flops.** This figure shows the variant curves of different methods with computational consumption as the x-axis, demonstrating the efficiency of our method.

## F  Evaluation on Image Diversity

We follow [24] to adopt an evaluation on the CLIP latent space, which is more in accordance with the T2I task. The metric *CLIP-Div*, as formulated below, with CLIP encoder $f_\theta$ and $k$ samples $\{\mathbf{x}_0^i\}_{i=1}^k$, calculates the difference between samples with the same prompt and directly measures the diversity on text-aligned features.

$$\text{CLIP-Div}\left(\{\mathbf{x}_0^i\}_{i=1}^k\right) := \sum_{i=1}^k \sum_{j=i}^k \frac{2}{k(k-1)} \left\| f_\theta(\mathbf{x}_0^i) - f_\theta(\mathbf{x}_0^j) \right\|_2^2 \tag{4}$$

Though the metric is under a relatively small range, we can judge the diversity by comparing the relative differences between different methods, as shown in Table 10. Generally, with the growing $N$, the generated results gain a higher possibility of convergence to adjacent superior features, resulting in a lower CLIP-Div score. However, the difference is quite small, indicating that our method enhances the performance with little sacrifice of diversity.

| Method | $N = 1$ | $N = 2$ | $N = 4$ | $N = 8$ |
|---|---|---|---|---|
| Importance Sampling | 0.0828 | 0.0819 | 0.0822 | 0.0794 |
| Best-of-N | 0.0828 | 0.0815 | 0.0788 | 0.0791 |
| Ours | 0.0792 | 0.0788 | 0.0789 | 0.0779 |

Table 10: **CLIP-Div** scores under different sampling strategies and values of $N$.

# G Ablation Study

## G.1 Pipeline Ablation

We present the ablation study of different design components in the overall pipeline in Table 11. As adaptive batch sampling alone (without integrated sample selection mechanisms) cannot directly enhance generation performance, these cases are denoted by "-". Excluding these baseline cases, both clustering-based diversity search and resampling-based potential selection demonstrate performance improvements, with statistically significant gains observed in reward and related evaluation indicators.

Notably, the clustering approach yields relatively moderate improvements, which can be attributed to its primary function of maintaining structural diversity rather than actively identifying superior samples for subsequent generation. The combination of diversity maintenance through clustering and quality-based selection via resampling synergistically enhances the effectiveness of the pipeline. This dual-mechanism framework ultimately achieves substantial performance gains over the baseline system, with the resampling component playing the pivotal role in selecting high-quality candidates for iterative refinement.

| $N$ | Method | GenEval | ImageReward | HPS | CLIP | Aesthetic |
|---|---|---|---|---|---|---|
| 2 | Infinity | 0.6946 | 1.1320 | 0.3042 | 0.3366 | 0.5811 |
| | +BoN | 0.7087 | 1.2545 | 0.3069 | 0.3384 | 0.5813 |
| | +Adaptive Batch Sampling | - | - | - | - | - |
| | +Clustering-Based Diversity Search | 0.7220 | 1.2591 | 0.3072 | 0.3385 | 0.5816 |
| | +Resampling-Based Potential Selection | 0.7403 | 1.4136 | 0.3106 | 0.3411 | 0.5821 |
| 4 | Infinity | 0.6946 | 1.1320 | 0.3042 | 0.3366 | 0.5811 |
| | +BoN | 0.7244 | 1.3471 | 0.3083 | 0.3397 | 0.5820 |
| | +Adaptive Batch Sampling | - | - | - | - | - |
| | +Clustering-Based Diversity Search | 0.7294 | 1.3608 | 0.3095 | 0.3403 | 0.5824 |
| | +Resampling-Based Potential Selection | 0.7437 | 1.4605 | 0.3112 | 0.3414 | 0.5821 |

Table 11: **Pipeline Ablation.** This table shows gains from each design.

## G.2 Computation-Performance Tradeoff

We evaluate our method with $N = 2, 4$ on the **GPU Nvidia A800-SXM4-80GB**. Below is the ablation of different designs and corresponding gains in inference time and peak memory utilization. Note that we do not include image decoding time here, as this is related to VAE and image size, which is irrelevant to the test-time scaling method itself. Because of the upper boundary of the 80G memory, BoN with $N = 8$ requires two inferences and doubles the consumption.

| $N$ | Method | GenEval | ImageReward | Inference Time (s) | Peak Memory (GB) |
|---|---|---|---|---|---|
| 1 | Infinity | 0.6946 | 1.1320 | 1.15 | 20.34 |
| 2 | Infinity | 0.6946 | 1.1320 | | |
| | +BoN | 0.7087 | 1.2545 | 1.70 | 28.12 |
| | +Adaptive Batch Sampling | - | - | 2.20 | 28.20 |
| | +Clustering | 0.7220 | 1.2591 | 2.28 | 29.90 |
| | +Resampling (Ours) | 0.7403 | 1.4136 | 2.51 | 30.73 |
| 4 | Infinity | 0.6946 | 1.1320 | | |
| | +BoN | 0.7244 | 1.3471 | 2.84 | 43.68 |
| | +Adaptive Batch Sampling | - | - | 3.99 | 43.87 |
| | +Clustering | 0.7294 | 1.3608 | 4.12 | 45.57 |
| | +Resampling (Ours) | 0.7437 | 1.4605 | 4.59 | 46.40 |
| 8 | Infinity+BoN | 0.7364 | 1.4144 | 5.70 | 43.68 |

Table 12: **Computation-performance tradeoff** of different designs.

Our method takes less time and memory to achieve comparable or even superior performance, for example, BoN with $N = 4$ and ours with $N = 2$. This proves the effectiveness and efficiency of TTS-VAR.

### G.3 Reward Models

We implement comparisons on using different reward models to rate the intermediate images and calculate the potential scores (VALUE), including Aesthetic [57], ImageReward [52], HPSv2 [56], and HPS+ImageReward. Owing to different value ranges of HPS and ImageReward, for HPS+ImageReward, we first calculate scores using the two models separately, then softmax the values of each model into the range $[0, 1]$, and finally take the average as the potential scores.

As shown in Table 13, generally, each reward model motivates an increase in the corresponding metric. For instance, with $N = 4$, the Aesthetic model, the ImageReward model, and the HPS model achieve the highest scores in the associated indicators, respectively. Among different models, ImageReward promotes improvements more. Especially with $N = 2$, ImageReward demonstrates a clear lead in GenEval and even defeats HPS in HPS score. We attribute this to the ability to clearly distinguish between superior and inferior samples, along with scoring that better aligns with human preferences.

| N | Reward Model | GenEval | ImageReward | HPS | CLIP | Aesthetic |
|---|---|---|---|---|---|---|
| 2 | - | 0.7087 | 1.2545 | 0.3069 | 0.3384 | 0.5813 |
| 2 | Aesthetic | 0.6966 | 1.123 | 0.3054 | 0.3366 | **0.6004** |
| 2 | ImageReward | **0.7403** | **1.4136** | 0.3106 | **0.3411** | 0.5821 |
| 2 | HPS | 0.7135 | 1.2246 | 0.3102 | 0.3391 | 0.583 |
| 2 | HPS+ImageReward | 0.7238 | 1.3522 | 0.3088 | 0.3402 | 0.5824 |
| 4 | - | 0.7244 | 1.3471 | 0.3083 | 0.3397 | 0.5820 |
| 4 | Aesthetic | 0.6842 | 1.1172 | 0.3056 | 0.3363 | **0.6114** |
| 4 | ImageReward | **0.7437** | **1.4605** | 0.3112 | **0.3414** | 0.5821 |
| 4 | HPS | 0.7255 | 1.2812 | **0.3154** | 0.3402 | 0.5843 |
| 4 | HPS+ImageReward | 0.7413 | 1.4128 | 0.3101 | 0.3406 | 0.5818 |

Table 13: **Reward Model Ablation.** This table shows results using different models for the potential.

### G.4 Ablation on Clustering

We compare our clustering method with a random drop process in Table 14. As shown in the results, even though resampling is in place to help ensure image quality, the effectiveness of scaling is significantly diminished when clustering is removed. This shows that clustering serves as an effective selection mechanism.

| N | Method | GenEval | ImageReward | HPS | CLIP | Aesthetic |
|---|---|---|---|---|---|---|
| 1 | Infinity2B+Ours | 0.7253 | 1.3226 | 0.3084 | 0.3395 | 0.5822 |
|   | w/o Clustering | 0.7193 | 1.2740 | 0.3072 | 0.3391 | 0.5821 |
| 2 | Infinity2B+Ours | 0.7403 | 1.4136 | 0.3106 | 0.3411 | 0.5821 |
|   | w/o Clustering | 0.7271 | 1.3672 | 0.3094 | 0.3403 | 0.5821 |
| 4 | Infinity2B+Ours | 0.7437 | 1.4605 | 0.3112 | 0.3414 | 0.5821 |
|   | w/o Clustering | 0.7369 | 1.4323 | 0.3103 | 0.3409 | 0.5830 |

Table 14: **Ablation study on clustering**.

### G.5 $\lambda$ Setting

In Table 15, we exhibit the results using different temperature $\lambda$ in the resampling process with fixed clustering operations. Intuitively, higher temperature promotes the expression of intermediate states with higher potential scores and prevents superior samples. However, excessively high temperatures can also widen the gap between intermediate states with the highest scores and those with scores that are only slightly lower. This can directly inhibit the generation of these slightly lagging intermediate states, which may ultimately become the optimal results. As shown, though there is a steady increase in ImageReward, $\lambda = 10.0$ falls behind $\lambda = 5.0$ with $N = 4$ in GenEval.

| $N$ | Lambda | GenEval | ImageReward | HPS | CLIP | Aesthetic |
|---|---|---|---|---|---|---|
| 2 | - | 0.7087 | 1.2545 | 0.3069 | 0.3384 | 0.5813 |
| 2 | 0.1 | 0.7065 | 1.2458 | 0.3065 | 0.3387 | 0.5811 |
| 2 | 0.5 | 0.7210 | 1.3167 | 0.3080 | 0.3400 | 0.5819 |
| 2 | 1.0 | 0.7222 | 1.3459 | 0.3087 | 0.3403 | 0.5821 |
| 2 | 5.0 | 0.7361 | 1.4010 | 0.3101 | 0.3410 | 0.5821 |
| 2 | 10.0 | **0.7403** | **1.4136** | **0.3106** | **0.3411** | **0.5821** |
| 2 | 50.0 | 0.7350 | 1.4132 | 0.3103 | 0.3410 | 0.5818 |
| 2 | 100.0 | 0.7244 | 1.3663 | 0.3087 | 0.3401 | 0.5809 |
| 4 | - | 0.7244 | 1.3471 | 0.3083 | 0.3397 | 0.5820 |
| 4 | 0.1 | 0.7308 | 1.3576 | 0.3089 | 0.3400 | 0.5816 |
| 4 | 0.5 | 0.7347 | 1.3918 | 0.3094 | 0.3406 | 0.5819 |
| 4 | 1.0 | 0.7418 | 1.4097 | 0.3099 | 0.3407 | 0.5820 |
| 4 | 5.0 | 0.7465 | 1.4500 | 0.3108 | 0.3412 | 0.5820 |
| 4 | 10.0 | 0.7437 | 1.4605 | **0.3112** | **0.3414** | **0.5821** |
| 4 | 50.0 | **0.7489** | **1.4629** | 0.3112 | 0.3413 | 0.5818 |
| 4 | 100.0 | 0.7394 | 1.4283 | 0.3103 | 0.3408 | 0.5818 |

Table 15: $\lambda$ **Ablation.** This table shows results with different lambda values.

# H   More Visualization Results

We present pairs of results on GenEval in Fig. 9 to compare the quality of Infinity, Infinity-IS, Infinity-BoN, and Infinity-*TTS-VAR*. These cases are sampled from text prompts in GenEval, which include the single object, two objects, counting, colors, position, and color attributes. As shown, our method produces higher-quality samples for the single object, such as the airplane in the left second line, effectively avoiding the generation of artifacts. In two-object settings, *TTS-VAR* successfully distinguishes references to different objects and generates accurate outputs based on prompts. For example, in the right second line, it eliminates conceptual mixtures and object disappearances. As illustrated in lines 3-10 on the right side, *TTS-VAR* also excels at determining counting numbers, positional relationships, and color attributes. Notably, in the last right line, our method generates a counterintuitive "green carrot," demonstrating its ability to separate objects from their natural attributes.

# I   Discussion on the Efficiency of Investing in Different Architectures

The efficiency of scaling across different model architectures remains an open question. In this section, we share our perspective on the potential benefits and limitations of scaling alternative architectures, such as VAR, in comparison to diffusion models.

**First, different architectures possess distinct characteristics and advantages.** Diffusion models, with their progressive denoising process and ability to model continuous latent spaces, generally achieve superior generative quality. By contrast, the VAR architecture more closely resembles the structure of large language models, which may be more conducive to unifying generation and understanding tasks [62]. Thus, depending on the specific objective, scaling efforts targeted at a given architecture may yield complementary benefits.

**Second, we believe that current scaling strategies for diffusion models are still at an early stage.** In contrast to the significant improvements demonstrated by our approach over Best-of-N (BoN), the search strategy introduced in [22] provides only marginal improvements over Random Search. We hypothesize that the multi-step denoising process inherent to diffusion models substantially enlarges the search space, thereby necessitating more sophisticated methodological designs. As a result, it remains difficult, based on current evidence, to make a definitive judgment on which scaling strategy or architectural choice is superior.

# J   Societal Impact

When applied to VAR models, *TTS-VAR* enhances the alignment of generated images with textual descriptions, making the generation process more controllable and better suited to meet creative and production demands. However, we also acknowledge the potential for misuse of this method, which could lead to privacy and copyright concerns. Nonetheless, we believe that our in-depth research into the VAR generation process will help researchers gain a clearer understanding, advance studies on

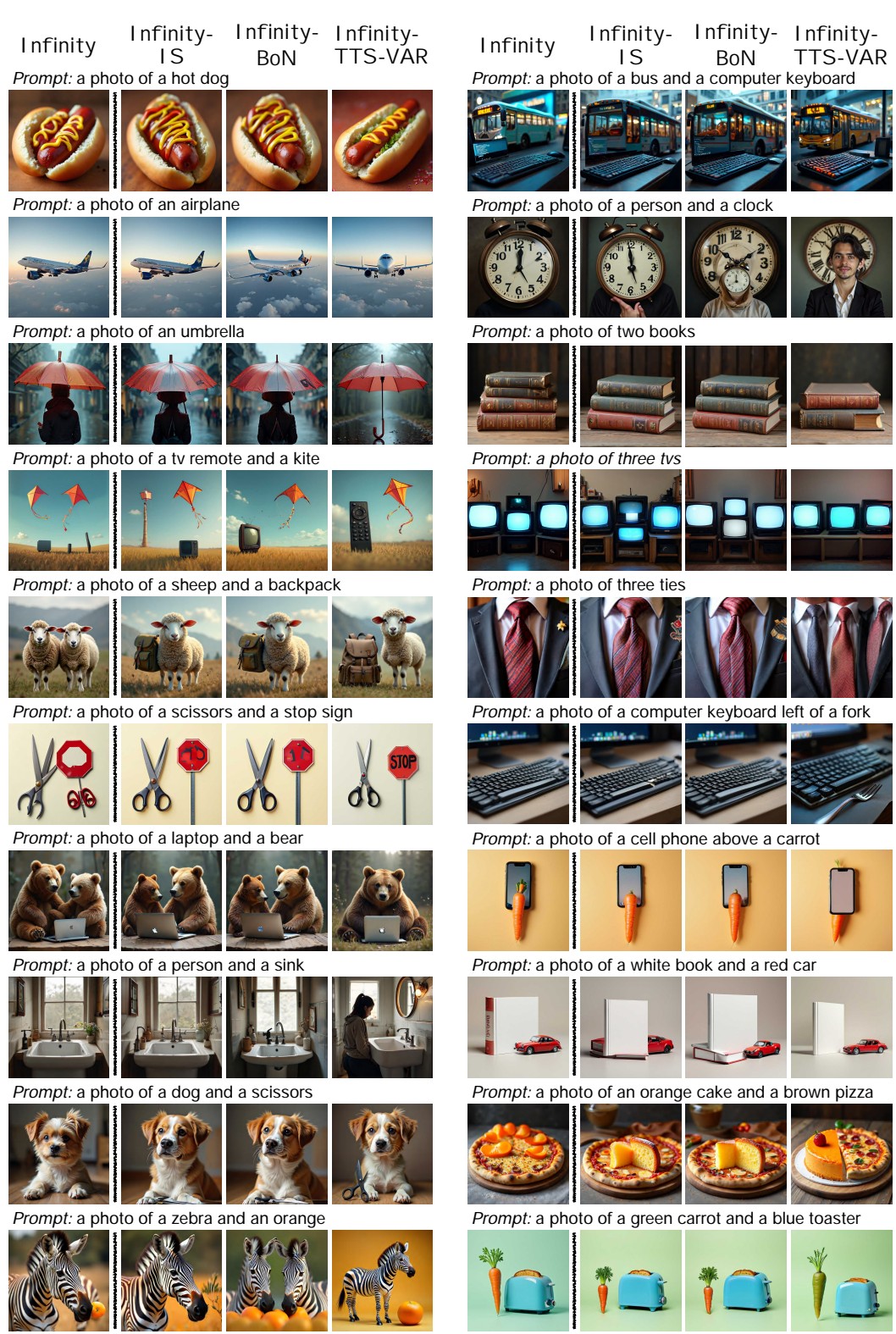

Figure 9: **More Visualization Results.** Samples are generated from GenEval prompts, with "IS" meaning Importance Sampling, "BoN" meaning Best-of-N, and our method *TTS-VAR*. We display various cases, including the single object, two objects, counting, colors, position, and color attributes.

controllability and safety in generation, and ultimately ensure that image generation models become safe and manageable tools.

## K   Limitation and Future Work

Though *TTS-VAR* shows significant improvement over the baseline and sets a new record, it still has two main limitations. First, *TTS-VAR* does not completely address the misalignment between text prompts and generated images. As indicated by the scores in Table 7, there are still some failure cases, particularly in the Position item. Second, while *TTS-VAR* is based on a general coarse-to-fine process, its potential application to other coarse-to-fine models, such as autoregressive models that use 1-D tokenizers, remains unexplored. In the future, we will investigate the generation process more thoroughly, examining the reasons for failure and designing solutions to unlock further scaling potential. Additionally, we plan to assess the compatibility of *TTS-VAR* with other autoregressive coarse-to-fine models, including those utilizing 1-D tokenizers and hybrid architectures that combine diffusion models. These efforts aim to create a more robust scaling framework for text-to-image synthesis while enhancing the methodological transferability of coarse-to-fine paradigms.

