# OpenReview forum: "TTS-VAR: A Test-Time Scaling Framework for Visual Auto-Regressive Generation"
_NeurIPS.cc/2025/Conference — NeurIPS 2025 poster_

### Official Review · Reviewer_QM2g · 2025-06-30

**Clarity:** 4
**Significance:** 2
**Originality:** 3
**Rating:** 4
**Confidence:** 5

**Summary:**

In the inference stage of VAR, this work employs semantic feature clustering in the early steps to maintain diversity in the sampling process. In the later steps, a reward model is used to compute potential scores and reorder candidate samples, thereby achieving computationally efficient test-time scaling.

**Questions:**

As the first TTS work for VAR architectures, I think it barely meets the acceptance criteria for NeurIPS, although I do have some concerns about this work (see weaknesses).

The authors adopt N=8 as the final setting, which requires starting inference with a batch size of 64. I am concerned that the performance improvement may be attributed to more sampling rather than the ingenuity of the proposed design.  Inference speed is also a factor that needs to be considered.

What is the performance when using only Clustering-Based Diversity Search during TTS?

**Ethical Concerns:**

["NO or VERY MINOR ethics concerns only"]

**Limitations:**

NO.
I do not find any potential negative societal impact in this work.

**Quality:**

3

**Strengths And Weaknesses:**

**Strengths**
1. This work is the first general test-time scaling framework for VAR models.
2. The proposed TTS-VAR achieves effective improvements on the GenEval benchmark.


**Weaknesses**
1. I am concerned about the inference speed of the model. This work starts with a batch size of 8*N and employs additional components such as DINOv2, k-means clustering, and multi-time reward model evaluations. Would these factors significantly reduce the inference speed? If we increase the value of N in the BoN, is it possible to achieve similar performance to TTS-VAR at a comparable inference speed?
2. Geneval reflects the semantic consistency of generated images under simple prompts and cannot capture the quality of details in the generated images. The advantages of high-scale resampling do not seem to be effectively captured by Geneval.
3. In Tab. 2, higher resampling frequency appears to lead to worse performance. Could the authors provide a more in-depth analysis of this phenomenon?

---

> ### Author Rebuttal · Authors · 2025-07-31
>
> We sincerely thank the reviewer QM2g for your insightful suggestions, and will add your advice to the camera-ready version. We appreciate your crucial questions, including inference efficiency, and we have provided analysis to address them below.
>
> &nbsp;
>
> ## **1. [W1] Computation-Performance Tradeoff of Different Designs**
>
> We evaluate our method with $N=2,4$ on the **GPU Nvidia A800-SXM4-80GB**. Below is the ablation of different designs and corresponding gains in inference time and peak memory utilization. Because the upper boundary of the 80G memory, BoN with $N=8$ requires two inferences and doubles the consumption.
>
> |$N$|Method|GenEval|ImageReward|Inference Time (s)|Peak Memory (GB)|
> |-|-|-|-|-|-|
> |1|Infinity|0.6946|1.1320|1.15|20.34|
> |||||||
> |2|Infinity|0.6946|1.1320|||
> ||+BoN|0.7087|1.2545|1.70|28.12|
> ||+Adaptive Batch Sampling|-|-|2.20|28.20|
> ||+Clustering|0.7220|1.2591|2.28|29.90|
> ||+Resampling (Ours)|0.7403|1.4136|2.51|30.73|
> |||||||
> |4|Infinity|0.6946|1.1320|||
> ||+BoN|0.7244|1.3471|2.84|43.68|
> ||+Adaptive Batch Sampling|-|-|3.99|43.87|
> ||+Clustering|0.7294|1.3608|4.12|45.57|
> ||+Resampling (Ours)|0.7437|1.4605|4.59|46.40|
> |||||||
> |8|Infinity+BoN|0.7364|1.4144|5.70|43.68|
>
> While these methods introduce some computational overhead, the impact is minimal. The adaptive batch size schedule, for instance, only adds a slight expense because it primarily increases the sample count when the sequences are short and the tokens are limited. Furthermore, by applying low-rank PCA (`pca_lowrank`) for dimensionality reduction before K-Means and utilizing a compact reward model on bf16, we effectively control filtering costs, ensuring that total resource consumption remains acceptable.
>
> Compared with BoN under similar time boundaries, for example, BoN with $N=4$ and Ours with $N=2$, there is a clear advance in GenEval Score, indicating the effectiveness and efficiency of our designs.
>
> &nbsp;
>
> ## **2. [W2] Other Benchmark Results**
>
> We acknowledge that GenEval's evaluation primarily focuses on text-image alignment and does not offer a comprehensive assessment of the overall generation quality. To provide a more complete result, in **Table 6** of the Appendix, we also report on metrics related to image quality, such as **ImageReward** and **HPS**, in addition to GenEval. These results are presented in the table below.
>
> |$N$|**Strategy**|GenEval|ImageReward|HPS|CLIP|Aesthetic|
> |-|-|-|-|-|-|-|
> |1|Raw Inference|0.6946|1.1320|0.3042|0.3366|0.5811|
> |1|Ours|**0.7253**|**1.3226**|**0.3084**|**0.3395**|**0.5822**|
> ||||||||
> |2|Importance Sampling|0.7022|1.1941|0.3051|0.3374|0.5807|
> |2|Best-of-N|0.7087|1.2545|0.3069|0.3384|0.5813|
> |2|Ours|**0.7403**|**1.4136**|**0.3106**|**0.3411**|**0.5821**|
> ||||||||
> |4|Importance Sampling|0.7116|1.2883|0.3067|0.3387|0.5815|
> |4|Best-of-N|0.7244|1.3471|0.3083|0.3397|0.5820|
> |4|Ours|**0.7437**|**1.4605**|**0.3112**|**0.3414**|**0.5821**|
> ||||||||
> |8|Importance Sampling|0.7181|1.3657|0.3085|0.3395|0.5810|
> |8|Best-of-N|0.7364|1.4144|0.3103|0.3406|**0.5820**|
> |8|Ours|**0.7530**|**1.4995**|**0.3122**|**0.3420**|0.5810|
>
> Furthermore, we have incorporated test results from **T2I-Compbench++** [1]. This benchmark assesses attributes such as color and shape, which, to some degree, reflect the quality of the generated images.
>
> |Model|Avg.|Color  (B-VQA)|Shape  (B-VQA)|Texture  (B-VQA)|2D Spatial  (UniDet)|3D Spatial  (UniDet)|Numeracy  (UniDet)|Non-spatial  (S-CoT)|Complex  (S-CoT)|
> |-|-|-|-|-|-|-|-|-|-|
> |Stable XL|0.5255|0.5879|0.4687|0.5299|0.2133|0.3566|0.4988|0.7673|0.7817|
> |Pixart-α-ft|0.5583|0.6690|0.4927|0.6477|0.2064|0.3901|0.5032|0.7747|0.7823|
> |Stable v3|0.6314|0.8132|0.5885|0.7334|0.3200|0.4084|0.6174|0.7782|0.7919|
> |DALLE· 3|0.6168|0.7785|0.6205|0.7036|0.2865|0.3744|0.5926|0.7853|0.7927|
> |FLUX.1|0.6087|0.7407|0.5718|0.6922|0.2863|0.3866|0.6185|0.7809|0.7927|
> |||||||||||
> |Infinity|0.5688|0.7421|0.4557|0.6034|0.2279|0.4023|0.5479|0.7820|0.7890|
> |Infinity2B+IS ($N=8$)|0.5965|0.7746|0.5078|0.6501|0.2462|0.4194|0.6002|0.7803|0.7937|
> |Infinity2B+BoN ($N=8$)|0.6115|0.7950|0.5439|0.6886|0.2545|0.4205|0.6090|0.7870|0.7937|
> |**Infinity2B+Ours ($N=2$)**|0.6151|0.7887|0.5578|0.6858|0.2697|0.4286|0.6112|0.7853|0.7936|
> |**Infinity2B+Ours ($N=8$)**|0.6230|0.8073|0.5914|0.7121|0.2644|0.4302|0.6340|0.7880|0.7963|
>
> We also introduced a user study on Google Forms, asking participants to select the best image from those generated by different methods based on three criteria: **Image Quality**, **Image Rationality**, and **Consistency with the Prompt**. This study consisted of 15 sample groups, and we collected 21 responses, resulting in 315 total votes. The percentages of votes received by each approach are summarized in the table below.
>
> In addition to receiving high marks for text consistency, these results also demonstrate that our method holds a competitive advantage over other approaches in terms of image quality and image rationality.
>
> |Metric|Baseline|IS|BoN|Ours|
> |-|-|-|-|-|
> |Image Quality|13.3%|7.9%|13.3%|65.4%|
> |Image Rationality|13.7%|8.6%|8.6%|69.2%|
> |Consistency with the Prompt|1.3%|1.9%|2.5%|94.3%|
>
> &nbsp;
>
> ## **3. [W3] Worse Performance with Higher Resampling Frequency**
>
> We discuss this phenomenon in Section 5.3 of the main submission. As shown in Figure 5, increasing the frequency of resampling does not always lead to performance improvements. This is because, as illustrated in Figure 5(b), the images decoded during the VAR generation process may not be effective representatives of the final generated image's quality. Consequently, resampling at early stages can lead to the erroneous selection of low-potential samples, thereby limiting the upper bound of scaling.
>
> &nbsp;
>
> ## **4. [Q1] Performance with only Clustering-Based Diversity Search**
>
> |$N$|Method|GenEval|ImageReward|HPS|CLIP|Aesthetic|
> |-|-|-|-|-|-|-|
> |2|Infinity|0.6946|1.1320|0.3042|0.3366|0.5811|
> ||+BoN|0.7087|1.2545|0.3069|0.3384|0.5813|
> ||+Adaptive Batch Sampling|-|-|-|-|-|
> ||**+Clustering**|0.7220|1.2591|0.3072|0.3385|0.5816|
> ||+Resampling (Ours)|0.7403|1.4136|0.3106|0.3411|0.5821|
> ||||||||
> |4|Infinity|0.6946|1.1320|0.3042|0.3366|0.5811|
> ||+BoN|0.7244|1.3471|0.3083|0.3397|0.5820|
> ||+Adaptive Batch Sampling|-|-|-|-|-|
> ||**+Clustering**|0.7294|1.3608|0.3095|0.3403|0.5824|
> ||+Resampling (Ours)|0.7437|1.4605|0.3112|0.3414|0.5821|
>
> In the table above, we present the ablation, including results of using only Clustering-Based Diversity Search for the cases where $N=2$ and $N=4$. Without resampling, we cannot filter intermediate states in the subsequent steps. Therefore, we cluster directly down to $N$ samples from the outset. Specifically, the adaptive batch size schedule used is {8**N**, 8**N**, 6**N**, 6**N**, 6**N**, 1**N**, 1**N**, 1**N**, 1**N**, 1**N**}.
>
> As shown, the clustering process effectively selects diverse intermediate states, ensures sample diversity, and provides more possibilities for the final selection, resulting in a significant improvement over the standard Best-of-N (BoN) method.
>
> &nbsp;
>
> **Reference**
>
> [1] T2I-CompBench++: An Enhanced and Comprehensive Benchmark for Compositional Text-to-image Generation. Huang, Duan, et al.

---

> ### Author Response · Authors · 2025-08-05
> **A Gentle Reminder of Feedback**
>
> Dear Reviewer QM2g,
>
> Thank you for your review and for acknowledging our work as the **first general TTS framework for VAR models** with **excellent clarity**.
>
> Following your insightful concerns, especially regarding efficiency, we have provided a detailed rebuttal. In summary, we have:
>
> - Added a direct comparison with Best-of-N under a **matched inference speed / compute budget**. The results confirm that our method's superiority comes from its efficient design, not just a larger sampling budget.
> - Included results on additional **qualitative metrics** to better demonstrate the improvement in image detail, addressing the limitations of GenEval.
> - Provided a deeper **analysis of the resampling frequency phenomenon** and included the requested **ablation study** on using the clustering alone.
>
> We believe these additions thoroughly address your concerns about the fairness and efficiency of our framework. As the discussion period is drawing to a close, we would be grateful to know if our response has resolved your questions.
>
> Thank you for your time and consideration.
>
> Sincerely,
>
> Authors of TTS-VAR

---

> ### Comment · Reviewer_QM2g · 2025-08-07
>
> Thanks for providing additional details for the paper. Most of my concerns are addressed by the authors and I would like to maintain my score.

---

> > ### Author Response · Authors · 2025-08-07
> >
> > Thank you for the positive feedback. We are glad that our response addressed most of your concerns. We commit to incorporating all promised changes in the final version.

---

### Official Review · Reviewer_4YxR · 2025-07-01

**Clarity:** 3
**Significance:** 2
**Originality:** 2
**Rating:** 4
**Confidence:** 4

**Summary:**

The paper introduces TTS-VAR, a novel test-time scaling framework for Visual Auto-Regressive (VAR) models, addressing the challenge of enhancing image generation quality without retraining. VAR models generate images hierarchically from coarse to fine scales, but early-stage tokens are hard to evaluate, leading to suboptimal sample selection. TTS-VAR tackles this via:
1. Clustering-based diversity search
2. Resampling-based potential selection
3. Adaptive descending batch size
Evaluated on the text-to-image VAR model *Infinity*, TTS-VAR improves the GenEval score by 8.7% (0.69→0.75) and outperforms diffusion/AR baselines with 60% fewer parameters.

**Questions:**

**Questions:**

- How does TTS-VAR framework affect the output distribution’s diversity? In [1], the paper demonstrates the FID and IS after scaling NFEs.
- What’s the computation-performance tradeoff of different designs?
- In table 3. and table 4., why does the proposed framework have limited improvements on HPS, CLIP and Aesthetic score?
- The efficiency of investing compute between diffusion models and VAR models?

Reference

[1] Inference-Time Scaling for Diffusion Models beyond Scaling Denoising Steps. Ma, Nanye, et al.

**Ethical Concerns:**

["NO or VERY MINOR ethics concerns only"]

**Final Justification:**

I thank the authors for their thorough and detailed rebuttal. They have addressed the main weaknesses pointed out in my initial review. My primary concerns were the lack of analysis on output diversity and the compute-performance trade-offs. The authors' response has resolved these issues, and I am maintaining my positive rating for the paper.

**Limitations:**

The limitation discussions are sound and clear to me.

**Paper Formatting Concerns:**

No such concerns.

**Quality:**

3

**Strengths And Weaknesses:**

**Strengths:**

- **Quality**:
    - Rigorous experiments across various metrics (GenEval, ImageReward, etc.) demonstrate consistent gains.
    - Ablation studies validate design choices (e.g., resampling scales, potential scores).
- **Clarity**:
    - Well-structured; figures (e.g., Fig 1, 5, 7) effectively illustrate the framework and scale-dependent behaviors.
    - Limitations and societal impact acknowledged (Sec 6, Appendix).
- **Significance**:
    - First general test-time scaling method for VAR models, enabling efficient high-quality generation.
    - Achieves SOTA results with lower computational costs (e.g., 25% samples vs. Best-of-N).
- **Originality**:
    - Novel integration of clustering (for structural diversity) and history-aware resampling tailored to VAR’s hierarchy.

**Weakness:**

- The paper does not provide quantitative or qualitative analysis of how the TTS-VAR framework affects output diversity. While the method emphasizes preserving structural diversity through clustering (Sec 4.2), there is no visualization comparing image variation between TTS-VAR and base Infinity model.
- The paper inadequately examines the compute-performance tradeoffs inherent in its design choices. Specifically, there is no analysis of how resampling at different scales impacts both computational cost (FLOPs, latency) and final output quality.

---

> ### Author Rebuttal · Authors · 2025-07-31
>
> We appreciate the reviewer 4YxR’s constructive feedback and will add your advice to the camera-ready version. We are encouraged that you acknowledged the originality of our method and the rigorous experiments demonstrating consistent gains across various metrics. We answer your questions below.
>
> &nbsp;
>
> ## **1. [W1, Q1] Output distribution diversity**
>
> The concept of output diversity can be divided into two components. The first is the diversity of the final output results, which we term **Final Semantic Diversity**. The second is the structural diversity among the intermediate states during the inference process, which we refer to as **Intermediate Structural Diversity**.
>
> ### **1.1 Final Semantic Diversity**
>
> In reference [1], the authors test IS and FID scores under the Class-to-Image (C2I) task on ImageNet. Since IS is based on the classification of generated images, it is not suitable for the T2I task. In addition, FID requires the reference distribution of real images, which is rarely adopted in open-domain T2I tasks.
>
> Therefore, we follow [2] to adopt an evaluation on the CLIP latent space, which is more in accordance with the T2I task. The metric *CLIP-Div*, as formulated below, with CLIP encoder $f_\theta$ and $k$ samples ${\\{x_0^i\\}}_{i=1}^k$, calculates the difference between samples with the same prompt and directly measures the diversity on text-aligned features.
>
> $$ \\text{CLIP-Div} \\left( \\{\\mathbf{x}\_0^i\\}\_{i=1}^k \\right) := \\sum\_{i=1}^k \\sum\_{j=i}^k \\frac{2}{k(k-1)} \\left\\|f\_{\\theta}(\\mathbf{x}\_0^i) - f\_{\\theta}(\\mathbf{x}\_0^j) \\right\\|\_2^2 $$
>
> Though the metric is under a relatively small range, we can judge the diversity by comparing the relative differences between different methods, as shown in the table below. Generally, with the growing $N$, the generated results gain a higher possibility of convergence to adjacent superior features, resulting in a lower CLIP-Div score. However, the difference is quite small, indicating that our method enhances the performance with little sacrifice of diversity.
>
> |Method|$N=1$|$N=2$|$N=4$|$N=8$|
> |-|-|-|-|-|
> |Importance Sampling|0.0828|0.0819|0.0822|0.0794|
> |Best-of-N|0.0828|0.0815|0.0788|0.0791|
> |Ours|0.0792|0.0788|0.0789|0.0779|
>
> ### **1.2 Intermediate Structural Diversity**
>
> We have visualized the generation process and analyzed the role of clustering in filtering these structural features. We found that clustering effectively groups samples with similar structures, thereby selecting for different structural characteristics and increasing the likelihood that various samples advance to later filtering stages.
>
> Take the intermediate process for the prompt, "Three origami cranes in white, gold, and blue are perched on a black lacquered shelf above a blooming bonsai tree," as an example. During generation, issues such as incorrect numbers of cranes, repeated colors, or wrong color sequences may arise. We observed that clustering can efficiently group samples with similar flaws into one category while grouping the generally correct samples into others. This filtering preserves representative samples, ensures structural diversity, and maximizes the probability that a correct sample exists for the subsequent reward-based filtering.
>
> Due to the inability to submit images during the rebuttal phase, we will incorporate visualizations of this filtering process in the camera-ready version of the paper.
>
> &nbsp;
>
> ## **2. [W2, Q2] Computation-Performance Tradeoff of Different Designs**
>
> We evaluate our method with $N=2,4$ on the **GPU Nvidia A800-SXM4-80GB**. Below is the ablation of different designs and corresponding gains in inference time and peak memory utilization. Because the upper boundary of the 80G memory, BoN with $N=8$ requires two inferences and doubles the consumption.
>
> |$N$|Method|GenEval|ImageReward|Inference Time (s)|Peak Memory (GB)|
> |-|-|-|-|-|-|
> |1|Infinity|0.6946|1.1320|1.15|20.34|
> |||||||
> |2|Infinity|0.6946|1.1320|||
> ||+BoN|0.7087|1.2545|1.70|28.12|
> ||+Adaptive Batch Sampling|-|-|2.20|28.20|
> ||+Clustering|0.7220|1.2591|2.28|29.90|
> ||+Resampling (Ours)|0.7403|1.4136|2.51|30.73|
> |||||||
> |4|Infinity|0.6946|1.1320|||
> ||+BoN|0.7244|1.3471|2.84|43.68|
> ||+Adaptive Batch Sampling|-|-|3.99|43.87|
> ||+Clustering|0.7294|1.3608|4.12|45.57|
> ||+Resampling (Ours)|0.7437|1.4605|4.59|46.40|
> |||||||
> |8|Infinity+BoN|0.7364|1.4144|5.70|43.68|
>
> Our method takes less time and memory to achieve comparable or even superior performance, for example, BoN with $N=4$ and ours with $N=2$. This proves the effectiveness and efficiency of TTS-VAR.
>
> Building upon Table 3 in the main submission, we have also included an analysis of the computation-performance tradeoff when applying resampling at different scales. As the results show, increasing the number of resampling steps incurs additional inference costs.
>
> |$N$|**Resampling Scale**|Inference Time (s)|GenEval|ImageReward|HPS|CLIP|Aesthetic|
> |-|-|-|-|-|-|-|-|
> |1|-|1.15|0.6946|1.132|0.3042|0.3366|0.5811|
> |||||||||
> |2|[6, 9]|1.78|**0.7133**|1.2572|0.3066|0.3379|0.5801|
> |2|[6, 8, 10]|1.84|0.7130|**1.2591**|0.3066|**0.3381**|0.5809|
> |2|[6, 7, 8, 9, 10, 11]|1.94|0.7114|1.2497|**0.3067**|0.3378|**0.5810**|
> |||||||||
> |4|[6, 9]|3.03|**0.7276**|1.3534|0.3082|**0.3398**|0.5817|
> |4|[6, 8, 10]|3.14|0.7247|**1.3592**|**0.3085**|0.3397|0.5822|
> |4|[6, 7, 8, 9, 10, 11]|3.33|0.7210|1.3558|0.3083|0.3398|**0.5830**|
>
> &nbsp;
>
> ## **3. [Q3] Limited Improvements on Other Indicators**
>
> In our view, these limited improvements can be attributed to two main factors.
>
> 1. **Different metrics possess inherent biases due to their distinct training data.** Although most metrics (excluding Aesthetic) evaluate text-image alignment to some degree, they have different tendencies and priorities. This is evidenced by the inconsistent rates of improvement across various metrics. In the majority of our experiments, we exclusively used ImageReward to guide the selection process. This method, however, primarily favors cases that align with the preferences of the ImageReward model itself. Consequently, while we observed notable improvements on GenEval, which focuses purely on text-image consistency, the gains on other metrics with specific biases, such as HPS and CLIP, were less pronounced.
>
> 2. **The metrics exhibit varying levels of numerical sensitivity.** As shown in Table 9 of the Appendix, even when using HPS as the reward for selection, the corresponding improvement in the HPS metric itself is not substantial. This is because, while all metrics are designed on the principle that a higher score indicates better quality, their specific numerical ranges and the magnitude of improvement represented by a score increase are not uniform across metrics.
>
> We also introduced a user study on Google Forms, asking participants to select the best image from those generated by different methods based on three criteria: **Image Quality**, **Image Rationality**, and **Consistency with the Prompt**. This study consisted of 15 sample groups, and we collected 21 responses, resulting in 315 total votes. The percentages of votes received by each approach are summarized in the table below.
>
> As shown, though some metrics did not show significant improvements, the results of the user study keenly capture the enhanced performance of our method across all three criteria, particularly in terms of consistency with the prompt.
>
> |Metric|Baseline|IS|BoN|Ours|
> |-|-|-|-|-|
> |Image Quality|13.3%|7.9%|13.3%|65.4%|
> |Image Rationality|13.7%|8.6%|8.6%|69.2%|
> |Consistency with the Prompt|1.3%|1.9%|2.5%|94.3%|
>
> &nbsp;
>
> ## **4. [Q4] Efficiency of Investing in Different Architectures**
>
> Overall, the benefits of scaling on different model architectures are not yet clearly established. Our work aims to provide a feasible solution for the alternative architecture VAR, and we are willing to share our perspective on this matter.
>
> 1. **Different architectures possess distinct characteristics and advantages.** Diffusion models, benefiting from their progressive denoising process and modeling of continuous latent spaces, tend to achieve superior generative quality. In contrast, the VAR architecture more closely resembles that of large language models, which may be more conducive to unifying generation and understanding, like [3]. Depending on the specific goal, scaling efforts on a particular architecture might yield distinct advantages.
>
> 2. **We believe that current scaling strategies for diffusion models are still in their early stages.** In contrast to the significant improvements our method shows over Best-of-N (BoN), the search method used in reference [1] offers only a marginal improvement over Random Search. We hypothesize that this is because the multi-step denoising process in diffusion models creates a much larger search space, likely requiring more sophisticated methodological designs. Therefore, based on the current works, it is difficult to make a direct judgment on which approach is definitively superior.
>
> &nbsp;
>
> **Reference**
>
> [1] Inference-Time Scaling for Diffusion Models beyond Scaling Denoising Steps. Ma, Nanye, et al.
>
> [2] A General Framework for Inference-time Scaling and Steering of Diffusion Models. Singhai, Horvitz, et al.
>
> [3] VARGPT-v1.1: Improve Visual Autoregressive Large Unified Model via Iterative Instruction Tuning and Reinforcement Learning. Zhuang, Xie, et al.

---

> ### Author Response · Authors · 2025-08-05
> **A Gentle Reminder of Feedback**
>
> Dear Reviewer 4YxR,
>
> Thank you for your thorough review and positive feedback. We are very pleased you recognized our work's **rigorous experiments**, **clarity**, and **novelty**.
>
> Following your excellent questions, we have provided a detailed rebuttal. In summary, we have:
>
> - Added a **quantitative analysis of output diversity** and an analysis of visualizations, as you suggested.
> - Included a **detailed compute-performance tradeoff analysis** for our key design choices.
> - Provided further discussion on the behavior of different evaluation metrics and efficiency between diffusion and VAR.
>
> We believe these additions directly address your main questions. As the discussion period is drawing to a close, we would be grateful to know if our response has resolved your questions.
>
> Thank you for your time and consideration.
>
> Sincerely,
>
> Authors of TTS-VAR

---

> > ### Comment · Reviewer_4YxR · 2025-08-07
> >
> > Thank the author for the replies to address my concerns. Thus, I will maintain my original positive rating.

---

> > > ### Author Response · Authors · 2025-08-07
> > >
> > > We are very grateful for your feedback and for your time in reviewing our manuscript. It is encouraging to know that our revisions and responses were satisfactory. We will add your suggestions to the camera-ready version.

---

### Official Review · Reviewer_XWDr · 2025-07-02

**Clarity:** 1
**Significance:** 3
**Originality:** 2
**Rating:** 4
**Confidence:** 3

**Summary:**

This paper presents TTS-VAR, a test-time scaling framework designed for visual autoregressive models by formulating generation as a path search task. To maintain computational efficiency, the authors introduce an adaptive descending batch size schedule, which operates similarly to an ensemble method with pruning to achieve a balance between performance and cost. For selecting high-quality candidate images, the method proposes a clustering-based diversity search to preserve structural information at coarse scales, followed by a reward model-based potential resampling to refine details at finer scales. Experimental results demonstrate that TTS-VAR improves generation quality.

**Questions:**

1. I suggest the authors include a detailed table comparing the time/memory cost of the method against each variant and baseline. Since this paper focuses on testing-time scaling behaviors, such a comparison is necessary.
2. Why use "clustering" instead of "pruning" or "selection"? The choice seems somewhat counter-intuitive.

**Ethical Concerns:**

["NO or VERY MINOR ethics concerns only"]

**Final Justification:**

The authors did provide additional clarifications and the extensive new experiments in the rebuttal, and most of my concerns have been addressed. I am raising my rating to **borderline accept**.

**Limitations:**

yes

**Quality:**

2

**Strengths And Weaknesses:**

### Strengths:

The authors make the first attempt to explore test-time scaling for visual autoregressive models to improve performance, which appears to be an interesting and promising direction.


### Weaknesses:

1. **Contribution of adaptive batch sampling:**

    In Fig. 2, I suggest the authors include additional curves showing memory usage and computational complexity with increasing scale index while using fixed but larger batch sizes (e.g., 2, 4, 8) for comparison. The current figure only illustrates the quadratic growth in computation due to increasing tokens but does not clarify the dependency on batch size. This weakens the motivation for adaptive batch sampling and clustering-based search.

2. **Comparison with Best-of-N (BoN):**

    The comparison in Fig. 1 between (b) and (c) seems unfair. BoN operates on a batch size of 2, while TTS-VAR starts with 8 initial samples, followed by clustering and resampling. Intuitively, BoN with resampling (batch size = 8) should perform at least as well as the current method, if not better, since no filtering is required. If a fixed batch size of 8 is feasible, why constrain the approach with decreasing batch sizes? The authors should provide a deeper discussion of this alternative, clarify how their method excels in efficiency, and present quantitative metrics to better evaluate the trade-offs.

3. **Design of clustering-based diversity search and resampling-based potential selection:**
    - The clustering approach needs further justification. Why does proximity to a cluster center imply high-quality structure? Additionally, clustering with very few samples (e.g., <8) may not be meaningful. What happens if all candidate samples are of poor quality? More visualizations (e.g., feature comparisons between superior and inferior samples, clustering results of "diverse" outputs) would help clarify this.
    - The reward model is used for resampling, but as noted in Supp. D.2, this overlaps with the model used for quantitative evaluation, potentially biasing the results. Leveraging this prior likely inflates performance metrics unfairly.
4. **Missing experimental results:**
    - Ablation of clustering-based diversity search: To validate clustering’s efficacy, the authors should include a baseline where N=8 samples are randomly dropped instead of clustered. The current experiments are insufficient.
    - While most experiments follow the setting of Infinity, results on the DPG benchmark are missing.
    - In Supp. Tab. 10, the authors should test λ > 10 to demonstrate that the current parameter choice is optimal.
5. **Confusing notations:**
    - L108: Replace i with k.
    - L158: The subscript of the union symbol should be j, not *bᵢ*.
    - L137-138: The definition of *N* is unclear and seems inconsistent with the batch size notation in evaluations (e.g., Tab. 1).

---

> ### Author Rebuttal · Authors · 2025-07-31
>
> We thank Reviewer XWDr for the insightful feedback. We will incorporate the suggestions in the camera-ready version and appreciate the recognition of our work. Our responses follow.
>
> &nbsp;
>
> ## **1. [W1] Contribution of Adaptive Batch Sampling**
>
> To illustrate the motivation for adaptive batch sampling, we compare it with fixed batch sizes below. While our method has higher initial FLOPs and memory usage, its computational growth is limited, and peak memory does not exceed that of a fixed batch size of 1. This design expands the search space at a minimal cost and prevents out-of-memory (OOM) errors.
>
> - FLOPs (TFLOPS)
>
> |Method|scale1|2|3|4|5|6|7|8|9|10|11|12|13|
> |-|-|-|-|-|-|-|-|-|-|-|-|-|-|
> |**Fixed 1**|0.0062|0.0298|0.1244|0.3370|0.7151|1.566|3.078|5.441|8.843|14.89|24.34|37.95|62.15|
> |**Fixed 2**|0.0120|0.0600|0.2490|0.6740|1.430|3.132|6.156|10.88|17.69|29.79|48.69|75.91|124.3|
> |**Fixed 4**|0.0246|0.1193|0.4975|1.348|2.861|6.263|12.31|21.76|35.37|59.57|97.38|151.8|248.6|
> |**Adaptive**|0.0493|0.2386|0.9949|2.271|4.540|9.643|15.69|20.42|27.22|39.32|48.77|62.38|86.58|
>
> - Memory (GB)
>
> |Method|scale1|2|3|4|5|6|7|8|9|10|11|12|13|
> |-|-|-|-|-|-|-|-|-|-|-|-|-|-|
> |**Fixed 1**|7.40|7.40|7.43|7.45|7.51|7.85|8.20|8.18|8.65|11.29|10.88|12.87|16.31|
> |**Fixed 2**|7.41|7.43|7.48|7.49|7.63|8.22|9.06|8.98|9.94|15.18|14.55|18.65|25.12|
> |**Fixed 4**|7.42|7.44|7.52|7.62|7.83|9.04|10.62|10.59|12.46|23.09|22.09|29.98|42.89|
> |**Adaptive**|7.45|7.48|7.71|7.76|8.06|9.84|10.59|8.98|9.96|15.20|11.06|12.99|16.45|
>
> We will incorporate the curves of other fixed batch sizes into the figure in the camera-ready version.
>
> &nbsp;
>
> ## **2. [W2, Q1] Comparison with BoN & Computation-Performance Tradeoff**
>
> 1. **The VRAM limit is a critical bottleneck in test-time scaling, especially in VAR models with quadratic memory growth.** A key challenge, therefore, is how to effectively expand the search space under severe memory constraints, which is also the primary motivation for our design of Adaptive Batch Sampling. While a BoN approach with $N=8$ should, in theory, perform greater than or equal to our method with $N=1$, it requires ~8x the VRAM. In practice, this leads to Out of Memory (OOM) errors even on an 80GB GPU (we used multi-GPU parallelism for our $N=8$ experiments).
>
> 2. **In terms of efficiency, our method's time overhead is minimal compared to increasing BoN's sample count.**  We tested the time and corresponding memory footprint for each component on the NVIDIA A800-SXM4-80GB GPU. Owing to the OOM error, BoN with $N=8$ requires two inferences and doubles the consumption. As shown, our method not only surpasses BoN with $N=8$ when ours is with $N=2$ in performance, but also shows a significant advantage in both time and memory consumption.
>
> |$N$|Method|GenEval|ImageReward|Inference Time (s)|Peak Memory (GB)|
> |-|-|-|-|-|-|
> |1|Infinity|0.6946|1.1320|1.15|20.34|
> |||||||
> |2|Infinity|0.6946|1.1320|||
> ||+BoN|0.7087|1.2545|1.70|28.12|
> ||+Adaptive Batch Sampling|-|-|2.20|28.20|
> ||+Clustering|0.7220|1.2591|2.28|29.90|
> ||+Resampling (Ours)|0.7403|1.4136|2.51|30.73|
> |||||||
> |4|Infinity|0.6946|1.1320|||
> ||+BoN|0.7244|1.3471|2.84|43.68|
> ||+Adaptive Batch Sampling|-|-|3.99|43.87|
> ||+Clustering|0.7294|1.3608|4.12|45.57|
> ||+Resampling (Ours)|0.7437|1.4605|4.59|46.40|
> |||||||
> |8|Infinity+BoN|0.7364|1.4144|5.70|43.68|
>
> &nbsp;
>
> ## **3. [W3] Design Discussion**
>
> ### **3.1 Effectiveness of Clustering**
>
> **The primary objective of clustering is not to find higher-quality structures, but to preserve the diversity of crucial structural features**, especially when the batch size is reduced due to VRAM limits. This approach ensures that a broader range of possibilities is available for evaluation in the subsequent resampling stage, enabling the selection of the most promising samples from a more diverse pool.
>
> Take the intermediate process for the prompt, "Three origami cranes in white, gold, and blue are perched on a black lacquered shelf above a blooming bonsai tree," as an example. During generation, issues such as incorrect numbers of cranes, repeated colors, or wrong color sequences may arise. We observed that clustering can efficiently group samples with similar flaws into one category while grouping the generally correct samples into others. This filtering preserves representative samples, ensures structural diversity, and maximizes the probability that a correct sample exists for the subsequent reward-based filtering.
>
> Due to the inability to submit images during the rebuttal phase, we will incorporate visualizations of this filtering process in the camera-ready version of the paper.
>
> ### **3.2 Using Reward Model as Metric Indicator**
>
> We acknowledge that using the reward score itself as an evaluation metric can indeed lead to a biased assessment. This is why we stated at the beginning of Section 5 that we use GenEval, which is an objective metric without bias on a specific indicator, as our main metric. We also report a detailed breakdown of its sub-metrics in the Appendix. For a more comprehensive evaluation, we also present results on another mainstream benchmark, **T2I-CompBench++** [1].
>
> |Model|Avg.|Color  (B-VQA)|Shape  (B-VQA)|Texture  (B-VQA)|2D Spatial  (UniDet)|3D Spatial  (UniDet)|Numeracy  (UniDet)|Non-spatial  (S-CoT)|Complex  (S-CoT)|
> |-|-|-|-|-|-|-|-|-|-|
> |Stable XL|0.5255|0.5879|0.4687|0.5299|0.2133|0.3566|0.4988|0.7673|0.7817|
> |Pixart-α-ft|0.5583|0.6690|0.4927|0.6477|0.2064|0.3901|0.5032|0.7747|0.7823|
> |Stable v3|0.6314|0.8132|0.5885|0.7334|0.3200|0.4084|0.6174|0.7782|0.7919|
> |DALLE· 3|0.6168|0.7785|0.6205|0.7036|0.2865|0.3744|0.5926|0.7853|0.7927|
> |FLUX.1|0.6087|0.7407|0.5718|0.6922|0.2863|0.3866|0.6185|0.7809|0.7927|
> |||||||||||
> |Infinity|0.5688|0.7421|0.4557|0.6034|0.2279|0.4023|0.5479|0.7820|0.7890|
> |Infinity2B+IS ($N=8$)|0.5965|0.7746|0.5078|0.6501|0.2462|0.4194|0.6002|0.7803|0.7937|
> |Infinity2B+BoN ($N=8$)|0.6115|0.7950|0.5439|0.6886|0.2545|0.4205|0.6090|0.7870|0.7937|
> |**Infinity2B+Ours ($N=2$)**|0.6151|0.7887|0.5578|0.6858|0.2697|0.4286|0.6112|0.7853|0.7936|
> |**Infinity2B+Ours ($N=8$)**|0.6230|0.8073|0.5914|0.7121|0.2644|0.4302|0.6340|0.7880|0.7963|
>
> &nbsp;
>
> ## **4. [W4] Other Experimental Results**
>
> ### **4.1 Ablation on Clustering**
>
> We compare our clustering method with a random drop process in the table below. As shown in the table below, even though resampling is in place to help ensure image quality, the effectiveness of scaling is significantly diminished when clustering is removed. This shows clustering is an effective selection mechanism.
>
> |$N$|Method|GenEval|ImageReward|HPS|CLIP|Aesthetic|
> |-|-|-|-|-|-|-|
> |1|Infinity2B+Ours|0.7253|1.3226|0.3084|0.3395|0.5822|
> ||w/o Clustering|0.7193|1.2740|0.3072|0.3391|0.5821|
> |2|Infinity2B+Ours|0.7403|1.4136|0.3106|0.3411|0.5821|
> ||w/o Clustering|0.7271|1.3672|0.3094|0.3403|0.5821|
> |4|Infinity2B+Ours|0.7437|1.4605|0.3112|0.3414|0.5821|
> ||w/o Clustering|0.7369|1.4323|0.3103|0.3409|0.5830|
>
> ### **4.2 Results on DPG-Bench**
>
> We evaluate the performance of the baseline Infinity model, the Best-of-N (BoN) method, and our approach on the DPG-Bench, specifically for the case where $N=4$. For each prompt, we generated four corresponding images for evaluation using seeds 0 through 3. The results are presented in the table below.
>
> |Methods|Global|Relation|Overall|
> |-|-|-|-|
> |SDv2.1|77.67|80.72|68.09|
> |DALL-E 3|90.97|90.58|83.50|
> |SDXL|83.27|86.76|74.65|
> |PixArt-Sigma|86.89|86.59|80.54|
> |SD3 (d=24)|-|-|84.08|
> |||||
> |HART|-|-|80.89|
> |Show-o|-|-|67.48|
> |Emu3|-|-|81.60|
> |||||
> |Infinity|80.96|89.83|81.72|
> |Infinity2B+BoN ($N=4$)|88.02|87.33|82.52|
> |**Infinity2B+Ours ($N=4$)**|83.37|88.81|82.94|
>
> ### **4.3 $\lambda$ > 10 Results**
>
> We conduct an ablation study on $\lambda \in \{50, 100\}$ for both $N=2$ and $N=4$, with the complete results presented in the table below. The rows labeled $\lambda=-$ represent the baseline results from BoN.
>
> A higher $\lambda$ promotes selecting better intermediate states, but an excessive value (e.g., 100) can hinder performance by prematurely pruning potential samples. The optimal setting varies with $N$: $\lambda=10$ performs best for $N=2$, whereas $\lambda=50$ achieves slightly better results for $N=4$.
>
> |$N$|Lambda|GenEval|ImageReward|HPS|CLIP|Aesthetic|
> |-|-|-|-|-|-|-|
> |2|-|0.7087|1.2545|0.3069|0.3384|0.5813|
> |2|1.0|0.7222|1.3459|0.3087|0.3403|0.5821|
> |2|5.0|0.7361|1.4010|0.3101|0.3410|0.5821|
> |2|10.0|**0.7403**|**1.4136**|**0.3106**|**0.3411**|**0.5821**|
> |2|50.0|0.7350|1.4132|0.3103|0.3410|0.5818|
> |2|100.0|0.7244|1.3663|0.3087|0.3401|0.5809|
> ||||||||
> |4|-|0.7244|1.3471|0.3083|0.3397|0.5820|
> |4|1.0|0.7418|1.4097|0.3099|0.3407|0.5820|
> |4|5.0|0.7465|1.4500|0.3108|0.3412|0.5820|
> |4|10.0|0.7437|1.4605|**0.3112**|**0.3414**|**0.5821**|
> |4|50.0|**0.7489**|**1.4629**|0.3112|0.3413|0.5818|
> |4|100.0|0.7394|1.4283|0.3103|0.3408|0.5818|
>
> &nbsp;
>
> ## **5. [W5, Q2] Notations and Words**
>
> We apologize for the incorrect notations on lines L108 and L158 and will correct them in the camera-ready version. Regarding the variable $N$ on lines L137-138, it represents the same as the $N$ in Table 1—the final number of generated samples. We will add a clarification on lines L137-138 to prevent any confusion.
>
> As for using clustering instead of pruning and selection, this decision is based on our observation of structural features that emerge early in the generation process. We found that these features can be effectively extracted, and by using clustering, we can preserve a diverse set of representative structures. This, in turn, provides a richer pool of candidates for the subsequent selection phase. Furthermore, direct pruning and selection are difficult at these early stages due to the lack of consistency between the early intermediate images and the final outputs. For these reasons, we chose and named our design 'clustering'.
>
> &nbsp;
>
> **Reference**
>
> [1] T2I-CompBench++: An Enhanced and Comprehensive Benchmark for Compositional Text-to-image Generation. Huang, Duan, et al.

---

> > ### Comment · Reviewer_XWDr · 2025-08-06
> >
> > I appreciate the authors' additional clarifications and the extensive new experiments, particularly the motivation behind adaptive batch sampling and clustering. Please ensure that these results, e.g. the analysis of compute-performance tradeoffs for adaptive batch sampling and further ablation studies, are included in the camera-ready version. Additionally, I have a suggestion for **Rebuttal 1 [W1]**: the two tables should include the *average* FLOPs and memory usage across all steps, as this would provide a clearer measure of the overall computational overhead.
> >
> > Given these improvements, I am raising my rating to **borderline accept**.

---

> ### Author Response · Authors · 2025-08-05
> **A Gentle Reminder of Feedback**
>
> Dear Reviewer XWDr,
>
> Thank you for your highly detailed and critical review. We are encouraged that you recognized our work as an "**interesting and promising direction**," and we have worked extensively to address the numerous concerns you raised.
>
> In our rebuttal, we have made substantial additions to address your concerns. In summary, we have:
>
> - **Strengthened our core motivation and comparisons:** We've added compute-performance tradeoffs and new analyses to justify our adaptive approach better.
> - **Clarified and justified key designs:** We provided deeper justification for our clustering method and discussed the potential evaluation bias from the reward model.
> - **Added extensive results:** We reported more ablation studies and results on the DPG benchmark.
>
> We hope these substantial revisions have addressed the core issues you identified. As the discussion period is drawing to a close, we would be very grateful to know if our response and revisions have helped clarify our contributions or if you have any further questions.
>
> Thank you for your time and consideration.
>
> Sincerely,
>
> Authors of TTS-VAR

---

> ### Author Response · Authors · 2025-08-06
> **Reply to Reviewer XWDr**
>
> Thank you for your positive feedback and for raising your score. We are glad to see that our rebuttal has addressed your questions. As promised, we will incorporate the improvements, including extra indicators of average FLOPs and memory, in the updated version. Your suggestions have helped us improve the clarity and presentation of our work.

---

### Official Review · Reviewer_wUiv · 2025-07-02

**Clarity:** 3
**Significance:** 3
**Originality:** 2
**Rating:** 5
**Confidence:** 3

**Summary:**

The article introduces a novel test-time scaling framework (TTS-VAR) for autoregressive models in the vision domain. The method proposed is able to take any trained text-to-image model and apply multi-scale path search algorithms and use the extra time to improve output quality by making batch sizes adaptive, employing clustering early for diversity and potential-based resampling once images are almost complete. Extensive empirical evidence suggests that TTS-VAR is able boost performance using few fewer parameters without touching the training process.

**Questions:**

1. Could you provide include more base-models on extended number of benchmark datasets to validate the generation of the proposed framework?
2. Would it be possible to provide concrete GPU times used on top of flops/memory usage to validate the practical improvements of TTS-VAR? Would be also nice to see how 60% deduction in parameters compared to SD-3 translate to real gains on inference efficiency (memory, time etc).
3. Could you provide a small set of human feedbacks on how TTS-VAR performs compared to baselines (such as best of n) on top of qualitative benchmarks?
4. Could you provide detailed hyperparameters used in the studies including but not limited to K (in K-mreans), seeds for experimentation and PCA energy threshold for reproducibility purposes?

**Ethical Concerns:**

["NO or VERY MINOR ethics concerns only"]

**Final Justification:**

The authors responses were extremely comprehensive and addressed most of my concerns.

**Quality:**

3

**Strengths And Weaknesses:**

Strengths:
1. Empirical gains are quite solid on GenEval 0.694 → 0.753 at N = 8 and already beats Best-of-N-8 with just N=2 while using 60% fewer parameters than SD-3.
2. The problem presented is well-motivated since test-time scaling has seen tremendous success in the text domain. The intuition for TTS-VAR makes sense and I think achieving SD-3 level quality without retraining could unlock a wide range of real-world applications with costs significantly reduced.
3. In general, the methodologies are clear to follow. Motivations for novel techniques such as adaptive batching, potential resampling and clustering are clearly motivated and formulated.
4. Ablations are comprehensive on the design choices of potential functions (table 3), resampling scales (table 2), clustering scales (table 4) and clustering extractors (table 5).

Concerns:
1. A problem I have with this paper is the lack of evaluation scope since only infinity-2B is used as the base model and GenEval as the sole benchmarking metric.
2. Figure 2 only presented Flop and memory information but does not include the type of GPU used and actual GPU hrs used to run the experiments.
3. There is limited disclosure of hyper parameters used in the studies including but not limited to K (in K-mreans), seeds for experimentation and PCA energy threshold.

---

> ### Author Rebuttal · Authors · 2025-07-31
>
> We sincerely thank the reviewer wUiv for positive feedback and insightful comments, and will add your advice in the camera-ready version. We are particularly encouraged that you found our work to be well-motivated and recognized its potential to unlock a wide range of real-world applications. We answer your questions below.
>
> &nbsp;
>
> ## **1. [W1, Q1] More Results for validation**
>
> ### **1.1 Other Model**
>
> Our test-time scaling framework is designed based on VAR-based Text-to-Image (T2I) models. Since Infinity is currently the only provider of an open-source T2I model with a VAR architecture, in addition to the 2B Infinity model used as base model in the paper, we selected another latest **8B model (resolution 512)** to validate the effectiveness of the framework. Specifically, since the generation process for the 512x512 resolution involves only 10 scales, we use an adaptive batch size schedule of {8**N**, 8**N**, 6**N**, 6**N**, 6**N**, 4**N**, 2**N**, 2**N**, 1**N**, 1**N**}. Clustering is applied to the scales in the range **[2, 5]**, and resampling is performed on the scales in the range **[6, 8]**.
>
> The experiments were conducted using the same settings as in Table 1 of the main submission and Table 6 in the Appendix. The results are shown in the table below.
>
> |$N$|**Strategy**|GenEval|ImageReward|HPS|CLIP|Aesthetic|
> |-|-|-|-|-|-|-|
> |1|Raw Inference|0.7646|1.2095|0.3081|0.3388|0.5721|
> |1|Ours|**0.7931**|**1.3929**|**0.3115**|**0.3414**|**0.5727**|
> ||||||||
> |2|Importance Sampling|0.7834|1.3091|0.3097|0.3399|0.5716|
> |2|Best-of-N|0.7880|1.3435|**0.3127**|**0.3423**|0.5722|
> |2|Ours|**0.7985**|**1.4572**|0.3106|0.3411|**0.5730**|
> ||||||||
> |4|Importance Sampling|0.7901|1.3786|0.3110|0.3409|0.5722|
> |4|Best-of-N|0.7995|1.4092|0.3122|0.3397|**0.5730**|
> |4|Ours|**0.8189**|**1.5100**|**0.3139**|**0.3425**|0.5727|
>
> As shown in the table, our framework also achieves excellent results on this model. Notably, when $N=4$, our method even surpasses the 0.80 threshold, delivering exceptional performance. We look forward to the emergence of more text-to-image models based on the VAR architecture, which will enable us to further validate our approach in the future.
>
> ### **1.2 Other Benchmarks**
>
> Recognizing the limitations of a single benchmark in fully assessing a method's effectiveness, we performed further evaluations on the **Infinity-2B** model using  **T2I-Compbench++** [1], a prominent benchmark in text-to-image generation. The corresponding results are detailed in the following table.
>
> |Model|Avg.|Color  (B-VQA)|Shape  (B-VQA)|Texture  (B-VQA)|2D Spatial  (UniDet)|3D Spatial  (UniDet)|Numeracy  (UniDet)|Non-spatial  (S-CoT)|Complex  (S-CoT)|
> |-|-|-|-|-|-|-|-|-|-|
> |Stable XL|0.5255|0.5879|0.4687|0.5299|0.2133|0.3566|0.4988|0.7673|0.7817|
> |Pixart-α-ft|0.5583|0.6690|0.4927|0.6477|0.2064|0.3901|0.5032|0.7747|0.7823|
> |Stable v3|0.6314|0.8132|0.5885|0.7334|0.3200|0.4084|0.6174|0.7782|0.7919|
> |DALLE· 3|0.6168|0.7785|0.6205|0.7036|0.2865|0.3744|0.5926|0.7853|0.7927|
> |FLUX.1|0.6087|0.7407|0.5718|0.6922|0.2863|0.3866|0.6185|0.7809|0.7927|
> |||||||||||
> |Infinity|0.5688|0.7421|0.4557|0.6034|0.2279|0.4023|0.5479|0.7820|0.7890|
> |Infinity2B+IS ($N=8$)|0.5965|0.7746|0.5078|0.6501|0.2462|0.4194|0.6002|0.7803|0.7937|
> |Infinity2B+BoN ($N=8$)|0.6115|0.7950|0.5439|0.6886|0.2545|0.4205|0.6090|0.7870|0.7937|
> |**Infinity2B+Ours ($N=2$)**|0.6151|0.7887|0.5578|0.6858|0.2697|0.4286|0.6112|0.7853|0.7936|
> |**Infinity2B+Ours ($N=8$)**|0.6230|0.8073|0.5914|0.7121|0.2644|0.4302|0.6340|0.7880|0.7963|
>
> As displayed, our approach yields substantial performance gains over the baseline Infinity model. Echoing the results on GenEval, our method with just $N=2$ outperforms BoN even when it uses $N=8$. While our method does not surpass SD3 on this benchmark, it significantly narrows the performance gap, achieving this result on a considerably smaller model.
>
> &nbsp;
>
> ## **2. [W2, Q2] Computation-Performance Tradeoff of Different Designs**
>
> We evaluate our method with $N=2,4$ on the **GPU Nvidia A800-SXM4-80GB**. Below is the ablation of different designs and corresponding gains in inference time and peak memory utilization. Because the upper boundary of the 80G memory, BoN with $N=8$ requires two inferences and doubles the consumption.
>
> |$N$|Method|GenEval|ImageReward|Inference Time (s)|Peak Memory (GB)|
> |-|-|-|-|-|-|
> |1|Infinity|0.6946|1.1320|1.15|20.34|
> |||||||
> |2|Infinity|0.6946|1.1320|||
> ||+BoN|0.7087|1.2545|1.70|28.12|
> ||+Adaptive Batch Sampling|-|-|2.20|28.20|
> ||+Clustering|0.7220|1.2591|2.28|29.90|
> ||+Resampling (Ours)|0.7403|1.4136|2.51|30.73|
> |||||||
> |4|Infinity|0.6946|1.1320|||
> ||+BoN|0.7244|1.3471|2.84|43.68|
> ||+Adaptive Batch Sampling|-|-|3.99|43.87|
> ||+Clustering|0.7294|1.3608|4.12|45.57|
> ||+Resampling (Ours)|0.7437|1.4605|4.59|46.40|
> |||||||
> |8|Infinity+BoN|0.7364|1.4144|5.70|43.68|
>
> Our method takes less time and memory to achieve comparable or even superior performance, for example, BoN with $N=4$ and ours with $N=2$. This proves the effectiveness and efficiency of TTS-VAR.
>
> |Model|Model Size|Resolution|Step (Scales)|Inference time (s)|Memory (GB)|GenEval|
> |-|-|-|-|-|-|-|
> |SD3 (d=38)|8B|1024$\times$ 1024|28|13.24|31.10|0.74|
> |Infinity+Ours ($N=2$)|2B|1024$\times$ 1024|13|2.51|30.73|0.7403|
>
> We display the comparison between Infinity+Ours ($N=2$) and SD3, because of similar memory consumption and GenEval score under the same resolution. As shown above, a 2B Infinity model with our test-time scaling achieves similar performance as an 8B SD3 model with less inference time (2.51s vs 13.24s).
>
> &nbsp;
>
> ## **3. [Q3] Human Feedback**
>
> To gather user feedback, we selected a small set of 15 image groups and compiled them into a questionnaire on Google Forms. For each group, the images are randomly shuffled, and participants were asked to select the best generated image based on three criteria: **Image Quality**, **Image Rationality**, and **Consistency with the Prompt**. In total, we collected 21 completed questionnaires, resulting in 315 sets of survey data for each indicator. The percentages of votes received by each approach are summarized in the table below.
>
> |Metric|Baseline|IS|BoN|Ours|
> |-|-|-|-|-|
> |Image Quality|13.3%|7.9%|13.3%|65.4%|
> |Image Rationality|13.7%|8.6%|8.6%|69.2%|
> |Consistency with the Prompt|1.3%|1.9%|2.5%|94.3%|
>
> As demonstrated, the results generated by our method received the most votes across all three criteria, highlighting its effectiveness, particularly regarding its consistency with the prompt.
>
> &nbsp;
>
> ## **4. [W3, Q4] Hyperparameters**
>
> In K-Means, we just set the next batch size $b_{i+1}$ in the adaptive descending batch sizes as the number of centers, to find $b_{i+1}$ different structure categories. For PCA, we only select the first major component, as we utilize it for dimension reduction only. In all benchmarks, we use seeds from `0` to `4` for different results. For experiments requiring multi-GPU, like $N=8$, we set the seed from `100*process_index` to `100*process_index+4` for each device.
>
> About other parameters related to Infinity, we just keep them the same as the original public settings.
>
> &nbsp;
>
> **Reference**
>
> [1] T2I-CompBench++: An Enhanced and Comprehensive Benchmark for Compositional Text-to-image Generation. Huang, Duan, et al.

---

> ### Author Response · Authors · 2025-08-05
> **A Gentle Reminder of Feedback**
>
> Dear Reviewer wUiv,
>
> Thank you once again for your constructive review. We are very encouraged that you recognized our work's **solid gains** and **clear methodology**.
>
> Following your valuable suggestions, we have updated our paper and provided a rebuttal. In summary, we have:
>
> - **Expanded our evaluation** with new model and benchmarks as you suggested.
> - Added **concrete compute-performance tradeoffs** and all **hyperparameters** for reproducibility.
> - Conducted a **human study** that confirms our method's qualitative advantage.
>
> We believe these additions have thoroughly addressed your concerns. As the discussion period is drawing to a close, we would be grateful to know if our response has resolved your questions.
>
> Thank you for your time and consideration.
>
> Sincerely,
>
> Authors of TTS-VAR

---

### Note · Authors · 2025-08-12

We would like to once again express our sincere gratitude to all the reviewers for their invaluable feedback. Our TTS-VAR, a pioneering test-time scaling framework for VAR models, improves generation quality and alignment using a novel multi-scale search and selection strategy.

We are encouraged by all reviewers' positive opinions, particularly their recognition of our work's novelty [*reviewer wUiv, 4YxR*], methodological clarity [*reviewer wUiv, 4YxR, QM2g*], and solid experimental results [*reviewer wUiv, 4YxR*]. The comments have also been instrumental in helping us to address the areas where our paper was lacking and to further enhance the quality of our work.

In our rebuttal, we have addressed the questions and concerns raised by the reviewers. To validate the robustness of our method [*reviewer wUiv, XWDr, QM2g*], we have incorporated additional models and benchmarks, and conducted a comprehensive user study. In terms of efficiency [*all reviewers*], we now include a detailed analysis of the performance-efficiency tradeoff for each component. Furthermore, we have added ablation studies [*reviewer XWDr, QM2g*] to further demonstrate the effectiveness of each part of our proposed design.

We hope our clarifications are helpful to both the reviewers and future readers. We are committed to further improving the paper for the camera-ready version.

---

### Decision · Program_Chairs · 2025-09-17

**Decision:**

Accept (poster)

**Comment:**

The final ratings for this paper are unanimously positive (one "Accept" and three "Boarderline Accept").

The AC agrees with the reviewers' positive evaluation.  The paper introduces TTS-VAR, a framework for test-time scaling of visual auto-regressive models. The main concerns shared by the reviewers are two folds. First, the initial experiments were limited in scope, focusing on a single base model and a primary benchmark, which raised questions about the method's generalizability. Second, for a paper focused on test-time efficiency, the initial draft lacked a detailed quantitative analysis of the compute-performance trade-offs.

The authors addressed these concerns in their rebuttal. They demonstrated the framework's effectiveness on a larger 8B model and on additional benchmarks (T2I-CompBench++, DPG-Bench). Also, they provided breakdowns of inference time and memory usage for each component, offering a detailed analysis of the efficiency trade-offs. They also supplemented their quantitative results with a user study. The authors' rebuttal have convincingly resolved the initial weaknesses identified by the reviewers.

Therefore, the AC recommends this paper for acceptance.